# Mitonuclear interactions shape both direct and parental effects of diet on fitness and involve a SNP in mitoribosomal 16s rRNA

**Adam J. Dobson** [1,2]*, **Susanne Voigt**[2], **Luisa Kumpitsch**[2], **Lucas Langer**[2], **Emmely Voigt**[2], **Rita Ibrahim**[1], **Damian K. Dowling**[3], **Klaus Reinhardt**[2]*

**1** School of Molecular Biosciences, University of Glasgow, Glasgow, United Kingdom, **2** Applied Zoology, Faculty of Biology, Technische Universität Dresden, Dresden, Germany, **3** School of Biological Sciences, Monash University, Melbourne, Australia

* adam.dobson@glasgow.ac.uk (AJD); klaus.reinhardt@tu-dresden.de (KR)

**Data Availability Statement:** Phenotype data and R code are available at github.com/dobdobby, and in supplementary materials. Sequence data are

## Abstract

Nutrition is a primary determinant of health, but responses to nutrition vary with genotype. Epistasis between mitochondrial and nuclear genomes may cause some of this variation, but which mitochondrial loci and nutrients participate in complex gene-by-gene-by-diet interactions? Furthermore, it remains unknown whether mitonuclear epistasis is involved only in the immediate responses to changes in diet, or whether mitonuclear genotype might modulate sensitivity to variation in parental nutrition, to shape intergenerational fitness responses. Here, in *Drosophila melanogaster*, we show that mitonuclear epistasis shapes fitness responses to variation in dietary lipids and amino acids. We also show that mitonuclear genotype modulates the parental effect of dietary lipid and amino acid variation on offspring fitness. Effect sizes for the interactions between diet, mitogenotype, and nucleogenotype were equal to or greater than the main effect of diet for some traits, suggesting that dietary impacts cannot be understood without first accounting for these interactions. Associating phenotype to mtDNA variation in a subset of populations implicated a C/T polymorphism in *mt:lrRNA*, which encodes the 16S rRNA of the mitochondrial ribosome. This association suggests that directionally different responses to dietary changes can result from variants on mtDNA that do not change protein coding sequence, dependent on epistatic interactions with variation in the nuclear genome.

## Introduction

Nutrition and genotype underpin variation in health and biological fitness. They can also interact, resulting in different responses among genotypes to the same nutritional changes [1,2]. In humans, there is interest in leveraging this variation to optimize nutrition by personalizing diet to individual consumers' needs [3]. To realize this ambition, we must understand the genetic drivers of variation in response to nutrition, but this is challenging because independently segregating loci have nonadditive, epistatic interactions [4], which may modulate responses to nutrition, i.e., diet-by-genotype-by-genotype variation [5]. Consequently, the genetic loci involved in these responses remain elusive.

available from NCBI SRA, accession
PRJNA853138.

**Funding:** This work was supported by a Dresden
Fellowship funded by the Excellence Initiative of the
German Federal and State Governments to A.D., a
UKRI Future Leaders Fellowship (MR/S033939/1)
to A.D., a University of Glasgow Lord Kelvin Adam
Smith Fellowship to A.D., and Deutsche
Forschungsgemeinschaft grant RE 1666/9-1 to K.
R. The funders had no role in study design, data
collection and analysis, decision to publish, or
preparation of the manuscript.

**Competing interests:** The authors have declared
that no competing interests exist.

**Abbreviations:** DMN, diet-by-mito-by-nuclear;
EAA, essential amino acid; EMM, estimated
marginal mean; GLM, generalized linear model;
GWAS, genome-wide association study; LD,
linkage disequilibrium; PCA, principal components
analysis; SNP, single nucleotide polymorphism.

Mitochondria are critical metabolic hubs, with their own small genome, and variation in their function can contribute to variation in dietary optima. The mitochondrial genome segregates independently of the nuclear genome, and the combination of mitochondrial and nuclear variants can generate "mitonuclear" epistasis [6,7]. This epistasis is thought to occur because mtDNA is transcribed, processed, and translated by nuclear-encoded proteins, and mtDNA-encoded proteins function in pathways and complexes that include nuclear-encoded proteins [8]. Reciprocally, outputs of genetic variation in the nucleus depend on how mitochondrial metabolites and signals feed into broader cellular networks. Mitonuclear epistasis has been reported for numerous traits and processes [9–11], but diet-by-mito-by-nuclear (DMN) interactions are less well characterized, despite evidence for mitochondrial modulation of nutrient signaling [12]. These interactions could be critical determinants of individual response to diet and, therefore, health. So far, DMN interactions have been shown for development time, life span, fecundity, and gene expression in *Drosophila melanogaster* [5,13–16]. This study addresses 4 main questions about DMN interactions, also using *D. melanogaster*: (A) How much phenotypic variation do DMN interactions cause, relative to lower-order interactions (i.e., nuclear-diet, mitochondria-diet, mitochondria-nuclear) and main effects (i.e., diet, nuclear, mitochondria)? (B) Parental nutrition can modulate offspring fitness, independent of offspring diet [17]—is this impact of nutritional variation modulated by mitonuclear variation? (C) Which specific dietary nutrients cause DMN variation? And, perhaps most importantly, (D) which mitochondrial polymorphisms underpin DMN interactions?

Here, in *Drosophila*, we study how variation among mitochondrial genotypes (mitogenotypes) modulates reproductive response to specific nutrients, in distinct populations of nuclear genotypes (nucleogenotypes). We identify variants in these genomes to characterise each population's specific combination of mitochondrial and nuclear variation (mitonucleogenotypes). We study reproductive traits because of their relevance to biological fitness, expanding on preceding studies [5,13–16] through multidimensional analysis of reproductive phenotype, and manipulating specific dietary nutrients (essential amino acids and lipid). We show that diets expected to promote fitness can in fact be lethal to specific mitonucleogenotypes. We also show that effects of parental nutrition on offspring performance are mitonucleogenotype specific. Effect sizes of DMN interactions were large for some traits, even exceeding those for diet:mitogenotype or diet:nucleogenotype interactions, implicating mitonuclear epistasis as a more important determinant of the response to nutrition than variation in either genome alone, and showing that DMN variation can be a major source of phenotypic variation. Importantly, we observe DMN interactions among a subset of populations differentiated only by an mtDNA polymorphism in a nonprotein-coding gene, long ribosomal RNA (*mt:lrRNA*), which encodes the mitoribosomal 16S rRNA. This gene is a structural component of the mitochondrial ribosome (mitoribosome) implicating a nonprotein-coding mitochondrial gene with roles in protein translation in DMN effects. Altogether, these results suggest that mitonuclear epistasis can be a leading determinant of optimal diet, that this variation maps to variants on mtDNA that do not change protein coding sequence, and that the consequences can be a matter of life or death.

## Results

### Establishing and sequencing a panel of diverse mitonucleogenotypes

Mitochondria are inherited exclusively from mothers through eggs. Mitonucleogenotype can therefore be manipulated by backcrossing virgin females of a given mitogenotype to males possessing a target nucleogenotype of interest and then iteratively crossing daughters produced by this cross to males with the target nucleogenotype across successive generations. This

procedure is expected to dilute and eventually purge the F0 mother's nucleogenotype in the mitonucleogenotype lineages produced, substituting it with the paternal nucleogenotype while retaining the F0 mother's mitogenotype. We used this approach to produce *D. melanogaster* populations with varied mitonucleogenotypes (Fig 1A), comprising replicated and fully factorial combinations of mitochondrial and nuclear genomes from Australia, Benin, and Canada (*A*, *B*, and *C*, respectively). With 45 females mating to 45 males in each iteration, the crossing scheme was designed to produce distinct mitochondrial backgrounds bearing equivalent pools of standing nuclear variation, by introgressing populations either reciprocally or to themselves. The use in F0 of multiple females from outbred ancestral populations maintained the potential for multiple mtDNA haplotypes to segregate in each of the introgressed populations. For brevity, we abbreviated population names, giving mitochondrial and then nuclear origin (e.g., *AB* = Australian mitochondria, Beninese nuclei). Each combination was triplicated (e.g., $AB_1$, $AB_2$, $AB_3$) at the beginning of the introgression, with triplicates maintained in parallel for more than 160 introgressions, altogether generating 27 populations, comprising 9 triplicated mitonucleogenotypes (Fig 1A) [18].

After >100 introgressions (S1 Table), assuming no mitochondrial incompatibility, we expected maternal nucleogenotypes to be purged by introgression and that nuclear variation among populations with co-originating nuclear backgrounds would be indistinguishable, regardless of mitogenotype. For a proof of principle, single nucleotide polymorphisms (SNPs) in the nuclear genome were identified by Pool-seq, sampling 2 populations per mito-nuclear pairing (i.e., 18/27 total). Within each nucleogenotype, principal components analysis (PCA) suggested negligible differentiation by mitogenotype, indeed samples with each nucleogenotype sat on top of one another in an ordination plot (Fig 1B), revealing 3 nucleogenotypes with no visible variation, on axes that explained 93% total variance (S1A Fig). For an orthogonal analysis of the grouping of nuclear genomes, we conducted continuous structure ("conStruct") analysis [19]. conStruct analysis produces a statistical model of population genetic structure, inferring similarity among a set of discrete genomes [19]. The method assigns genetic variation to a set of *K* possible user-specified states ("layers") in the model. Each sample is then depicted as a contribution from each layer, aka proportion admixture of the different hypothetical layers. We specified *K* = 3 possible layers to reflect 3 founder populations at the beginning of the introgression. The conStruct analysis complemented the PCA, assigning co-originating nuclear genomes almost entirely to the same layers (Fig 1C), and each individual population was assigned between 94.4% and 99.9% to its respective dominant layer (S2 Table). Together, the PCA and conStruct analyses indicated high degrees of similarity between co-originating nucleogenotypes, and differentiation between nucleogenotypes of different geographic origin.

We then examined among-mitogenotype population structure, reanalyzing previously reported mitochondrial Pool-seq data [18]. PCA (Fig 1D) revealed consistent within-mitogenotype clustering, except population $AA_3$, which was distinct from other *A* mitogenotypes, and intermediate between the *B* and *C* mitogenotypes on the first PC. Little differentiation was apparent between mitogenotypes *B* and *C* on the first PC. On the second PC, *C* mitogenotypes were equivalent to *A*. The majority of mitogenotype variance was explained by these 2 PCs (S1B Fig). The $AA_3$ mitogenotype was strikingly intermediate between the 2 major clusters of other mitogenotypes; however, this was not wholly surprising, given that (A) the ancestral population originated from the middle of a cline on which 2 main mitochondrial haplotypes have been reported [20] and that (B) *D. melanogaster* settled relatively recently in Australia, likely founded by both European and African lineages [21], which may have introduced haplotypes with the intermediate genotype we observed.

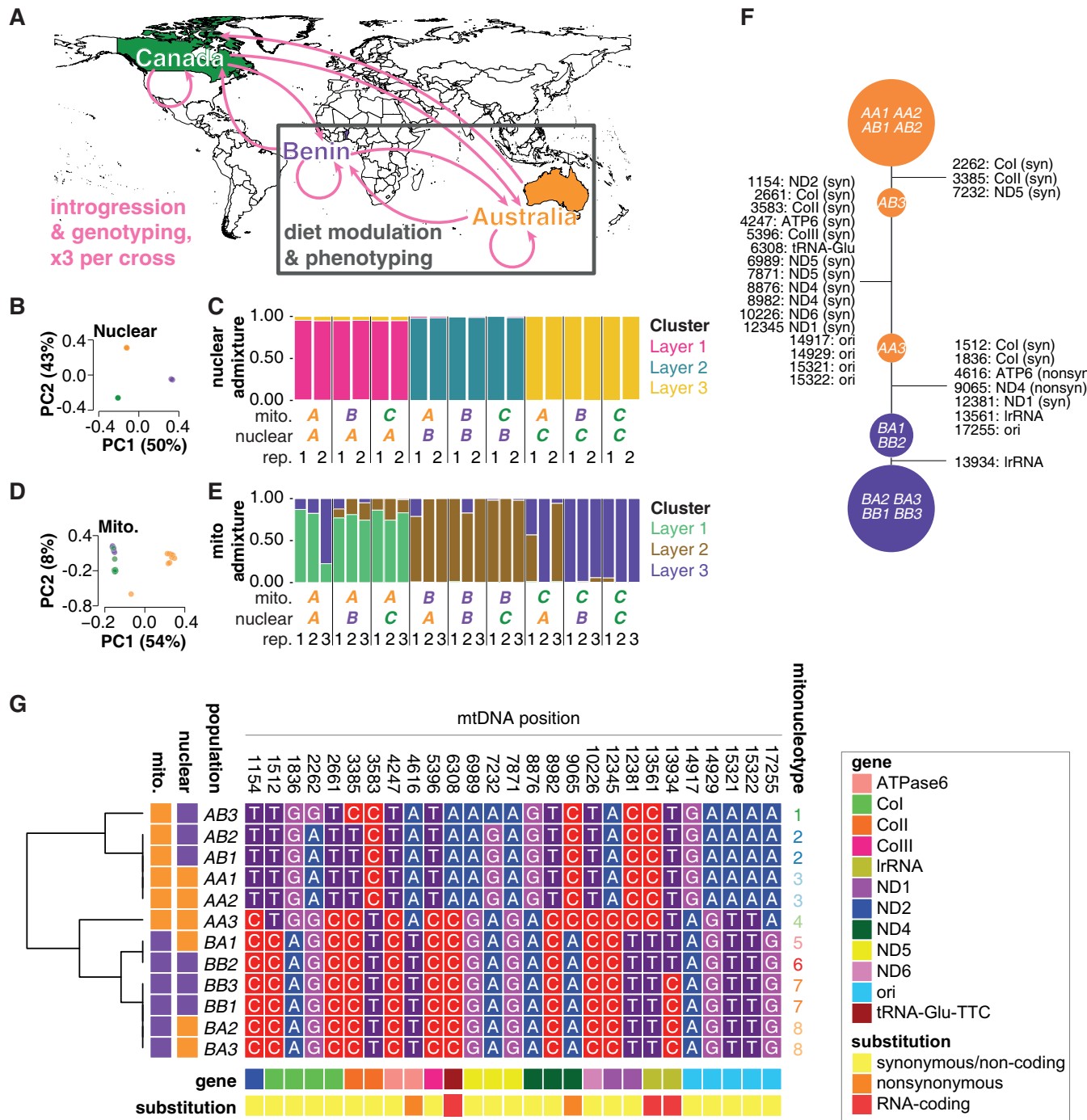

**Fig 1. A panel of diverse mitonucleogenotypes in *D. melanogaster*: Population setup and grouping according to mitochondrial and nuclear SNPs. (A)** Fly populations from Australia, Benin, and Canada were introgressed in all possible pairwise combinations, generating novel combinations of mitochondrial and nuclear genomes. Three biologically independent replicate populations were established per introgression. In every generation, 45 females and 45 males were crossed, allowing the potential for variation to segregate within each population. Map created in R with Natural Earth data. (**B**) PCA indicates purging of $F_0$ mothers' nucleogenotypes, and homogeneous substitution with nuclear genomes from donor populations. PCA was performed on per-population allele frequencies, of all observed nuclear SNPs on the major chromosome arms (2L, 2R, 3L, 3R, and X). Each point represents a distinct population, color-coded as per panel (A). Points representing diverse nucleogenotypes sit on top of one another, suggesting homogenized nuclear genomes even in the presence of distinct mitochondria. (**C**) Proportional assignment of populations to clusters ("layers") according to nuclear SNP frequency, by admixture analysis. Admixture proportions for each population were inferred by model-based clustering with ConStruct. Colors represent the proportion variants in each populations' genome assigned to arbitrary clusters ("layers"). The analysis was instructed to assign populations to 3 layers (K = 3), because we expected 3 major groupings resulting from the 3 distinct geographic origins. Most variation in genomes originating from Australia, Benin, and Canada was assigned to clusters/layers 1, 2,

and 3, respectively. To minimize effects of LD, only nuclear SNPs at least 1 kb apart and outside regions of no recombination were considered for PCA and admixture analyses. Population IDs are given below the barplot (mito. = mitogenotype, nuclear = nucleogenotype, rep = population replicate). Note that only 2 populations were sequenced per mito-nuclear combination, assuming that the anticipated nuclear homogenization would be equivalent in all 3 populations: The equivalent assignments of nucleogenotypes to layers suggest that this assumption was correct, and nuclear genotype is shaped by the nuclear intogression, independent of mitochondrial genotype, This recapitulates the PCA result (Panel B). (**D**) PCA shows 2 major groupings of mitogenotypes according to all observed mtDNA SNPs. Each point represents a distinct population, colour-coded as per panel (A). The intermediate population on PC1 represents population $AA_3$, suggesting a mitogenotype in this population that is intermediate between the major clusters comprising mitogenotypes A (to the right) and mitogenotypes B and C (to the left). (**E**) Mitochondrial admixture proportions, showing assignment of populations to layers according to mtDNA SNP frequency. Admixture proportions for each population were inferred by model-based clustering with ConStruct (K = 3). Colors represent proportion assigned to each layer. Most variation in mtDNA originating from Australia, Benin, and Canada was assigned to clusters/layers 1, 2, and 3, respectively. The result suggests high levels of similarity among replicate mitogenotypes, largely independent of nucleogenotype, i.e., recapitulating the PCA result (Panel D). (**F**) Network analysis based on the major alleles in each population at the 27 differentiated sites. Populations are grouped according to allele frequency at indicated loci on mtDNA. SNPs distinguishing each cluster of populations are indicated in text, showing mtDNA position, gene, and whether for protein-coding genes whether the SNP was synonymous or not. (**G**) Segregation of major alleles for significantly differentiated mitochondrial SNPs. Heatmap shows nucleotide identity at positions in mitochondrial genome indicated at top. Gene for each position and SNP classification (synonymous/nonsynonymous) indicated by color bar at top, and geographic origin of mitochondrial and nuclear genomes indicated on right. Hierarchical clustering (dendrogram on left) shows separation of SNPs by geographic origin, with 5 constituent clusters. Concatenating SNP clusters with nucleogenotype reveals 8 mitonucleogenotypes, indicated to right. Data underlying the graphs shown in the figure can be found in S11–S14 Tables. LD, linkage disequilibrium; PCA, principal components analysis; SNP, single nucleotide polymorphism.

We also conducted conStruct analysis on the mitochondrial genomes—although we interpret results conservatively because mtDNA SNPs are unlikely to segregate independently (due to the presence on this diminutive genome of only a few SNPs, in high linkage disequilibrium (LD) because of lack of recombination). Despite these caveats, the conStruct analysis complemented the PCA analysis, with populations $CA_1$ and $CA_3$ dominated by admixture with the Beninese mitogenotypes (Fig 1E) and population $AA_3$ appearing again as an outlier, intermediate between other mitogenotypes (Fig 1E). These findings suggested that phenotyping both *B* and *C* mitochondria in the same experiment would likely prove redundant, and so we decided to eliminate one of these mitogenotypes from the study, reducing the number of populations to be phenotyped, for experimental tractability. We retained mitochondria with Beninese origins because of this background's widespread use in fly research but excluded mitochondria with Canadian origins. We also excluded Canadian nucleogenotypes (*AC*, *BC*) because of lack of coevolutionary history with *A* or *B* mitogenotypes. This reduced number of populations for phenotyping from 27 to 12 (i.e., studying only $AA_{1-3}$, $AB_{1-3}$, $BA_{1-3}$, and $BB_{1-3}$). To confirm that this reduced panel contained DMN variation necessary for subsequent phenotyping, we conducted a preliminary phenotypic analysis, focusing on fecundity (i.e., egg laying/24 hours). We chose fecundity because of the relevance of reproductive traits to Darwinian fitness. We applied both an established dietary manipulation that promotes fecundity by enriching essential amino acids (EAAs) [22], and a novel manipulation that represses fecundity (S2A Fig), by enriching plant-based lipids (Text A in S1 Text). Feeding these EAA-enriched and lipid-enriched diets to the focal panel of 12 populations (S2B Fig and Text B in S1 Text) revealed mitonuclear variation in fecundity response (S2C Fig, Text B in S1 Text). This motivated further study of the specific SNPs that differentiated these populations, and how these SNPs predicted variation in a comprehensive analysis of reproductive phenotype.

Among the focal Australian and Beninese populations, we characterized mtDNA polymorphisms in detail to identify genetic information that could be used to model phenotypic responses to diet. We tested for SNPs at significantly different frequencies (Fisher's exact test, FDR < 0.001), finding 28 altogether (S3 Table), which were predominantly biallelic (S3 Fig). Positions of these SNPs on a map of the mitochondrial genome are shown in S4 Fig. Only one SNP was significantly differentiated between mitogenotypes *B* and *C*, which further validated our decision to exclude Canadian populations from our analysis. By contrast, 27 of the 28 SNPs were significantly differentiated between mitogenotypes *A* and *B* (S3 Table), i.e., the

majority of mitochondrial diversity in the full panel was represented by these 2 mitogenotypes. Even though these alleles could still potentially segregate, major alleles for the majority of SNPs (70%) were fixed, 89% of SNPs were at a frequency ≥0.99, and the lowest observed frequency for a SNP (position 17,255) was nevertheless still high at 0.8 (S3 Fig and S3 Table). Thus, allele frequency differed significantly among populations, but major alleles were at high frequency within each population.

Since the populations were not isogenic, and the backcrossing regime was designed to permit segregating variation in each genome, we expected mtDNA variation might segregate within individual populations, and so allele frequencies could potentially drift. We therefore resequenced mtDNAs at a 2-year interval, but the maintenance of allele frequency over these 2 years suggested that any drift was negligible (S5 Table) and, therefore, that the populations were a legitimate resource for genetic association studies.

We examined the known functions of mtDNA loci bearing SNPs. Of the 27 SNPs, 19 were in protein-coding regions, in genes encoding subunits of electron transport chain complexes and *ATPase subunit 6*. Only 2/19 were predicted to be nonsynonymous (S3 Table). Interestingly, one nonsynonymous C/A SNP, at mtDNA position 9,065, coding for a valine/leucine substitution in *ND4*, has previously been characterized, including enhanced sensitivity to a high-protein diet [23]. In our populations, it cosegregated with variation at multiple other positions and so cannot be characterized further here. We are not aware of previous reports of the other nonsynonymous variant, an A/T SNP at mtDNA 4,616, predicted to encode a methionine/isoleucine substitution in *ATPase subunit 6*. The remaining 17/19 SNPs in protein-coding sequence were predicted to be synonymous. The other 8/27 SNPs were in nonprotein-coding regions (in the origin of replication, a tRNA, and the mitoribosomal 16S rRNA, *lrRNA*) (S3 Table). Thus, altogether, the populations were differentiated by 27 mtDNA SNPs, 25 of which were predicted to not affect protein amino acid sequences.

We explored how these SNPs were distributed among the populations to identify groups for subsequent genetic associations. Pool-seq analysis identifies alleles, but we emphasize the potential by experimental design for variation to segregate in our populations and that we did not sequence haplotypes: Alleles of distinct SNPs may theoretically segregate independently of one another among our populations. We therefore studied the differentiation of populations by major allele cosegregation using a network analysis (Fig 1F). This reinforced findings of our previous analysis of the full set of populations (i.e., including Canadian mitogenotypes) [18]. The network revealed a punctuated continuum of among-population variation, independent of nucleogenotype. Some populations had unique mitogenotypes ($AB_3$, $AA_3$). Others had identical mitogenotypes even in the presence of distinct nucleogenotypes ($AA_1$, $AA_2$, $AB_1$, $AB_2$; and $BA_2$, $BA_3$, $BB_1$, $BB_3$; and $BA_1$, $BB_2$). Australian mitogenotypes and Beninese mitogenotypes were largely dichotomous, except that population $AA_3$ was notably intermediate (Fig 1F), consistent with the preceding PCA and admixture analyses. Thus, the mitonuclear populations could be grouped by frequencies of major alleles on mtDNA, revealing 4 groups of mitogenotypes.

How did mtDNA alleles intersect with nucleogenotypes—what mitonucleogenotypes were present in the panel? We clustered the populations by major alleles on mtDNA, using hierarchical clustering, and examined the intersection with nucleogenotype. Because our PCA and admixture analyses of nDNA suggested that co-originating nucleogenotypes were homogenized and shaped by introgression (i.e., not incompatibility with mtDNA) in this particular panel, we viewed nucleogenotype as a dichotomous factor (A or B) for this analysis. The clustering separated *A* mitogenotypes from *B* (Fig 1G). However, within this geographic differentiation, the sequencing revealed more granular among-population differentiation, with 5 distinct clusters of unique mitogenotypes. These mitogenotypes were not nested within

nucleogenotype, and some co-occurred with both *A* and *B* nucleogenotypes, which indicated that mitonuclear incompatibilities were not at play during the introgression process (consistent with PCA and admixture). To generate a final, sequence-informed mitonucleogenotype assignment, for genetic associations, we concatenated sequenced-based mitogenotype with nucleogenotype (i.e., A or B, since our sequencing data suggested nuclear homogenization independent of mitogenotype (Fig 1)). This revealed 8 distinct mitonucleogenotypes (Fig 1G). We then phenotyped the responses of each mitonucleogenotype to diet and examined how traits were shaped by mitonucleogenotype.

We noticed that a subset of populations (mitonucleogenotypes 5, 6, 7, and 8) (Fig 1G) were distinguished by only one mitochondrial polymorphism, suggesting that any mitonuclear or DMN variation in this subset was attributable to this SNP. This contrasted other populations, which bore confounding variation at other positions, and so effects could not be attributed to a single locus. The specific SNP was a C/T polymorphism in *mt:lrRNA* (mtDNA position 13934), which occurred at high frequencies (between 0.99 and 1; S3 Table) in each nuclear background. *mt:lrRNA* encodes the 16S RNA of the mitochondrial ribosome, which seemed like a good candidate to mediate mitonuclear effects, because (1) the proteins that this RNA forms complexes with in the mitochondrial ribosome (mitoribosome) are encoded by the nuclear genome; (2) the mitochondrial ribosome translates mitochondrial proteins, so variation in its function has the potential to generate a bioenergetic bottleneck, with metabolic consequences for the rest of the cell, and consequences for penetrance of nucleogenotype variation; and (3) preceding work found a role for a tRNA in mitonuclear effects [24]. We hypothesized that the C/T polymorphism in *mt:lrRNA* may provide an illustrative example of how nonprotein-coding variation in the mitochondrial genome could underpin DMN variation in phenotype. We therefore decided to investigate the effects of this SNP specifically in subsequent genetic mapping, alongside analyses of the full panel of populations.

## Phenotyping

Encouraged by initial fecundity results (S2 Text), we characterized a more extensive panel of fitness traits, examining how they responded to dietary variation and how those responses associated with mitonucleogenotype. Using a total of >25,000 individual flies, we assayed fecundity, fertility, and development time—traits that have previously been shown to be sensitive to DMN variation [13,16]—as well as number of adult progeny as a direct fitness measure. As with our preliminary investigation of fecundity, we fed flies either a control medium, an EAA-enriched medium that promotes egg laying, or a lipid-enriched medium that reduces egg laying (Texts A and B in S1 Text). Flies were maintained on a distinct "development" medium prior to experiments, before switching to experimental diets in adulthood (Fig 2B), so that all flies including controls experienced a novel diet upon switch to experimental food, to distribute any novelty effects evenly among conditions. Fig 2A shows approximate nutrient content of this diet, along with control diet, EAA-enriched diet, and lipid diet. We then varied feeding on these experimental diets in 2 different ways (Fig 2B). Fitness effects of long-term dietary changes for both parent and offspring are to be expected, and previous work [13] has shown that such changes can elicit DMN variation. However, diet can also influence offspring health when manipulated only in parents, independent of offspring diet [17,25–27]. To study whether such effects of parental diet are mitonucleogenotype dependent, we exposed flies to either a chronic feeding paradigm, in which both parents and offspring were fed experimental diets, or a parental feeding paradigm, in which diets were fed transiently to parents before eggs were laid and developed on a standardized medium, distinct from parental diet (Fig 2B). In the latter context, DMN interactions can only result from parental effects. To ensure genetic

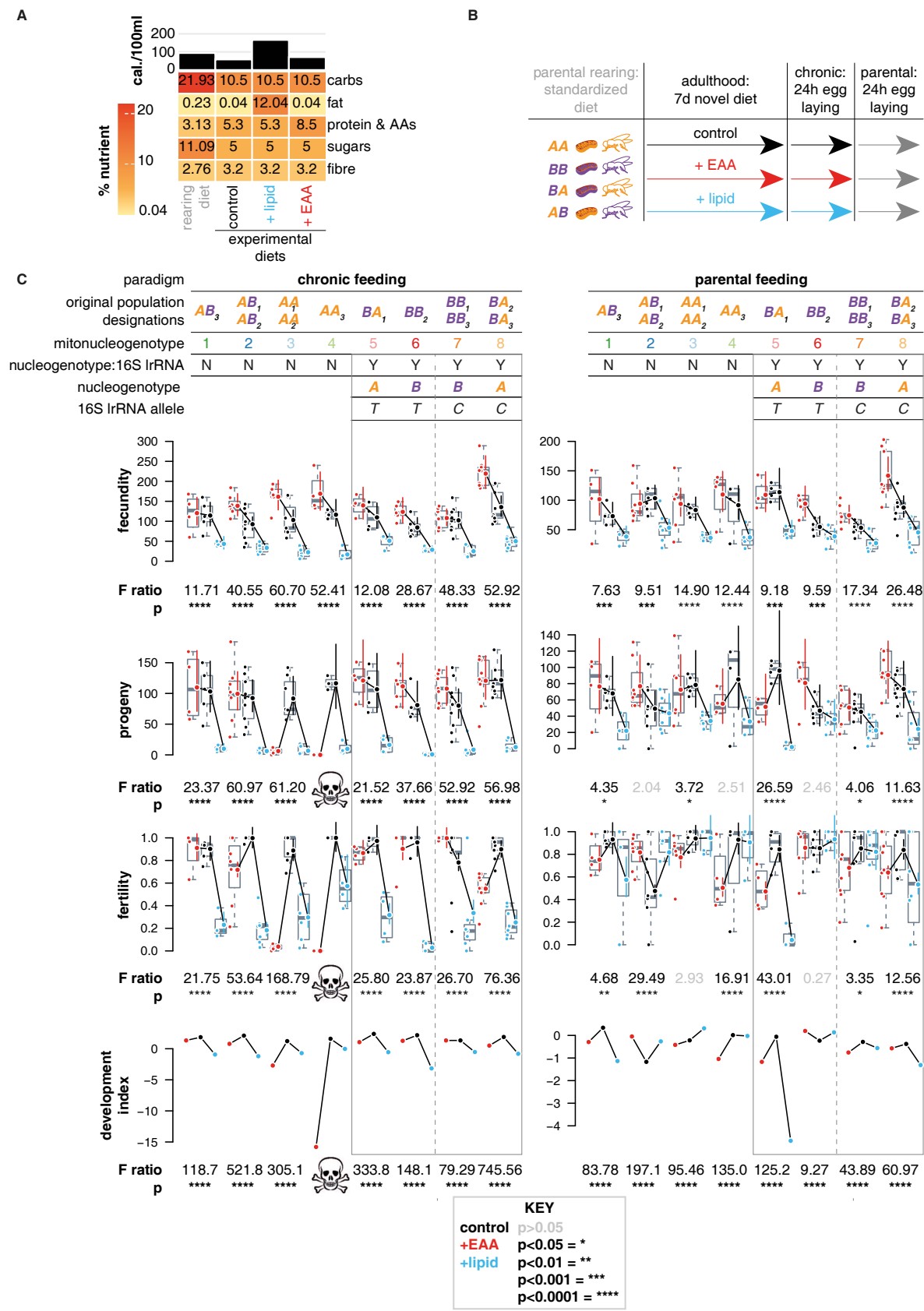

**Fig 2. Mitonucleogenotypes modulate multitrait responses to chronic and parental nutritional variation and show that a single C/T polymorphism in a subset of populations is sufficient to induce diet-mito-nuclear variation.** (**A**) Diet design: The heatmap shows estimated macronutrient content of diets used in this study; bars at top indicate caloric content. (**B**) Key and experimental design. Flies were reared from egg to adult on rearing food and allocated at random to experimental media 6–48 hours after eclosion, at a density of 5 of each sex per vial. After 7 days, flies laid eggs on fresh food for 24 hours, followed by a further 24 hours on standardized rearing medium. (**C**) Mitonuclear variation in response to chronic and parental changes in nutrition. Significant mitonucleogenotype:diet interactions were observed among the full set of populations, and significant diet:mito:nuclear interactions were observed for progeny, fertility, and fecundity in the subset of populations whose mitochondrial genomes were differentiated by only one SNP in *mt:lrRNA*. Top of the plot indicates feeding paradigm and original population designation (e.g., $AA_1$, $AA_2$), and mitonucleogenotype based on mtDNA sequence and nDNA origin (see Fig 1G). Top of plot also denotes subset of populations used to assess interaction between nucleogenotype and C/T polymorphism in *mt:lrRNA* (position 13934), indicated by "nucleogenotype:16S lrRNA" Y/N (Y = subset used for analysis), nucleogenotype, and the major allele at this SNP for each given mitonucleogenotype ("16S lrRNA allele"). Subset of populations analyzed for effect of *mt:lrRNA* are highlighted by a grey box, dashed lines down center of panels separate the T and C alleles. Panels below show estimated marginal means (EMMs) for trait indicated on y-axis, with error bars indicating 95% confidence intervals (note the confidence intervals are sometimes small, and eclipsed by the plot point). Colors encode diet as per key, egg and progeny counts are presented as *x+1* to enable plotting on log scale. Development index shows EMMs for Cox mixed-effects models of proportion eclosed over time, excluding sex from plot. Development data are plotted in full as Kaplan–Meier plots in S6 Fig. Statistics below each group of points give F-statistics and *P* values (Tukey corrected) for effect of diet in each given mitonucleogenotype, calculated by ANOVAs of each trait's full model, stratified per mitonucleogenotype using joint tests. Absence of diet effect in a given mitonucleogenotype (*p*-value > 0.05) is indicated by grey text. Data underlying the graphs shown in the figure can be found in S22–S24 Tables.

consistency, the same parents were used in each paradigm, by laying eggs for 24 hours after 1 week on experimental media (chronic paradigm), then switching to a universal standardized medium (the medium that the flies developed on) for another 24 hours of egg laying (parental paradigm).

We assessed how phenotypic variation in fecundity, progeny, fertility, and development partitioned by mitonucleogenotype and diet. No trait varied as a linear function of caloric density in any mitonucleogenotype (S5 Fig), and so diet was modeled as an unordered factor. To visualize variation, we calculated estimated marginal means (EMMs; [28]) with confidence intervals. For fecundity, progeny, and fertility, EMMs are statistical coefficients, approximating the trait values imputed to the model. For development, EMMs are a coefficient of a model representing both time to emergence and whether or not an egg developed to adulthood, integrating both parameters into a single development index. Importantly, the EMMs are calculated from statistical models, enabling visual comparisons among conditions, which was useful for our multicondition, multitrait study.

Plotting EMMs per mitonucleogenotype indicated considerable variation in response to diet (Fig 2C). Plotting by geographic origin of mitochondria and nuclei confirmed the same (S6B Fig). Statistical models revealed ubiquitous diet-mitonucleogenotype variation (generalized linear models (GLMs) for fecundity, progeny, and fertility; Cox models for development time), except for fecundity in the parental feeding paradigm (S6 Table). For development time models, we also included interactions with offspring sex, because of reports of sex-biased mito: nuclear variation [16]. However, sex did not modify diet:mitonucleogenotype interactions (all *p* > 0.05; S6 Table), suggesting that DMN effects on this trait were not sex biased in these populations. Chronic lipid feeding was deleterious for all traits, but mitonucleogenotype shaped magnitude. We were surprised that chronic EAA feeding promoted fecundity, across all populations, but suppressed fertility (Fig 2C), reducing progeny counts to below those of control diet, with the consequence that fitness was not ultimately not enhanced by EAAs. The magnitudes of changes induced by EAAs were mitonucleogenotype dependent (Fig 2C).

Mitonucleogenotypes 3 and 4 stood out in the chronic feeding paradigm, because their progeny counts after chronic EAA feeding were even lower than after chronic lipid enrichment, to near lethality in mitonucleogenotype 3 (Figs 2C and S6D). Again, this effect in mitonucleogenotype 4 was not observed if either mitochondria or nuclei were switched (mitonucleogenotypes 2 or 8), confirming another mitonuclear effect. Mitonuclear

incompatibility is widely reported [29], as are DMN effects on physiology and life history [5,13–16]: The present data now indicate that mitonuclear incompatibility can be diet dependent, under nutrient-enriched conditions that we had expected to promote fitness.

DMN variation was also apparent in the parental feeding paradigm, albeit less pronounced than after chronic feeding. To our knowledge, this is the first evidence that mitonucleogenotype modulates effects of parental nutrition. Lipid was less universally toxic upon parental feeding than chronic feeding, and mitonucleogenotypes 3 and 4 stood out, exhibiting a benefit of parental lipid feeding, developing on average 1 day earlier (S6D Fig). However, mitonucleogenotype 4 shared a mitogenotype with mitonucleogenotype 2, and a nucleogenotype with mitonucleogenotype 8, but neither mitonucleogenotypes 2 or 8 showed the same behavior, indicating that the fitness benefit of parental lipid feeding is a mitonuclear interaction effect. The mitonuclear variation in response to parental diet was not universal among all populations. Variation in response to parental diet was pervasive in offspring traits (i.e., progeny, fertility, development index). Not all populations showed statistically significant effects of parental diet, e.g., for number of progeny, these effects were restricted to mitonucleogenotypes 1, 3, 5, 7, and 8 (Fig 2C). These populations were not all the same as those that showed an effect of parental diet on fertility (1, 2, 4, 5, 7, and 8; Fig 2C). Thus, impacts of parental diet on offspring fitness appear to manifest at the level of integration between distinct traits, and mitonuclear variation means that not all populations respond to parental nutritional variation.

For an aggregate view of traits per mitonucleogenotype, we conducted a PCA of trait values (EMMs), which showed that mitonucleogenotype 4 had a response to EAA-enriched food that was distinct not only from mitonucleogenotype 3 but in fact distinct from all other populations in the experiment (S7 Fig). Our sequencing had shown that this population's mitogenotype was intermediate between other groups of populations at an mtDNA-wide level (Fig 1D), intermediate in the network of significantly differentiated SNPs (Fig 1F), and intermediate in the clustering of significantly differentiated SNPs (Fig 1G). Thus, this population's mitogenotype is atypical for either Australia or Benin (Fig 1G), with some loci bearing alleles at high frequency in other populations bearing Australian mitogenotypes, and other loci bearing alleles at high frequency in other populations bearing Beninese mitogenotypes. This population bore Australian nuclei, and its mitochondria originated from Australia but were clearly distinct from mitonucleogenotype 3, and its phenotype responded to diet differently. Therefore, we speculate that the lethality of our specific nutrient treatments indicates incompatibility between the Australian nuclear genome and mtDNA loci bearing Benin-like alleles. This incompatibility appears to be diet dependent in this population. We also noted that mitonucleogenotype 5 had a distinct response to high-lipid diet, showing a compromised development index (discussed below).

We applied statistical analysis to confirm diet:mitonucleogenotype effects. We excluded mitonucleogenotype 4 from some statistical analysis because its extreme trait values complicated modeling (see Texts A-D in S1 Text). Among the other populations, ANOVA tests revealed significant mitonucleogenotype:diet interactions (S6 Table). To estimate variability in response to dietary change, we calculated F-ratios and $P$ values for effect of diet per mitonucleogenotype (Fig 2C). F-ratios varied up to 10-fold, depending on trait (Fig 2C). Diet effects were significant for all mitonucleogenotypes in the chronic feeding paradigm ($p < 0.001$ in all cases) but not in the parental feeding paradigm. These analyses suggest that variance in response to diet can be partitioned by sequence-based mitonucleogenotype [22].

## Phenotyping *mt:lrRNA* SNP

We were particularly interested by the paucity of nonsynonymous mtDNA polymorphisms, which suggested that DMN effects may be underpinned by variation outside of protein-coding

regions. As detailed above, mitonucleogenotypes 5, 6, 7, and 8 bore fully factorial variation in nucleogenotype and the *mt*:*lrRNA* SNP. Plots of phenotypes in mitonucleogenotypes 5, 6, 7, and 8 (Fig 2C) revealed both quantitative variation in fertility effects of diet but also qualitative changes in the sign of the response to dietary change. Specifically, in populations with the *mt*:*lrRNA* T allele, chronic EAA feeding decreased fertility in both nucleogenotypes (mitonucleogenotypes 5 and 6). However, the *mt*:*lrRNA* C allele unleashed nucleogenotype-dependent responses to chronic diet: C allele populations with nucleogenotype *A* showed decreased fertility after EAA feeding (mitonucleogenotype 8) but increased fertility with nucleogenotype *B* (mitonucleogenotype 7) (Fig 2C). Indeed, mitonucleogenotype 7 was the only mitonucleogenotype that increased fertility upon chronic EAA feeding. In the parental feeding paradigm, nucleogenotypes *A* and *B* responded to diet equivalently in the presence of the *mt*:*lrRNA C* allele (mitonucleogenotypes 7 and 8). However, nucleogenotype-specific responses to parental diet were unleashed by the *mt*:*lrRNA* T allele: Fertility was impaired by parental feeding on either EAA or lipid in the presence of nucleogenotype *A* (mitonucleogenotype 5) but not in the presence of nucleogenotype *B* (mitonucleogenotype 6). This altered fertility had apparent consequences for progeny count and development index (Fig 2C). Statistical tests (S7 Table) confirmed interactions of the *mt*:*lrRNA* polymorphism, nucleogenotype, and diet, for all traits except egg laying. This exclusively postembryonic variation indicated impacts on offspring performance but not parental reproductive effort.

We also analyzed how the geographic origin of mitochondria and nuclear genome modulated the response to diet (Text C in S1 Text) because this allowed us to assess variance explained by mitochondria and nuclei separately (this is not possible when information is concatenated into mitonucleogenotype, and the fully factorial variation is required to fit a 3-way DMN interaction term). This analysis accorded with our sequence-based analysis of mitonucleogenotype (Text C in S1 Text), revealing DMN variation for all traits except fecundity in the parental paradigm, effects of lipid feeding, and effects of parental diet.

## Effect size calculations

Our final analysis assessed the extent to which mitonucleogenotype:diet interactions, and lrRNA:nucleogenotype:diet interactions, shaped phenotypic variation in each respective analysis. We calculated an estimate of effect size (partial $\eta^2$) that allowed us to compare impacts of predictive variables (Fig 3). We calculated this measure from test statistics [30] using the R *effectsize* library [31], deriving F values from post hoc EMM tests. (This method of calculating *partial* $\eta^2$ differs from $\eta^2$ in that the resulting values do not necessarily sum to 1.) We calculated partial $\eta^2$ for all traits, in both feeding paradigms, for each of the 3 different types of analyses we had conducted (i.e., sequence-driven mitonucleogenotype assignment (Fig 3A), the specific analysis of the SNP in *lrRNA* (Fig 3B), and geographic origin of populations (Fig 3C)). We compared the higher-order interactions we were interested in to lower-order effects, anticipating that diet would be the largest source of variation for most traits, but that this this might be modified by DMN interactions, mitonucleogenotype, or the *lrRNA*:nucleogenotype interaction. However, the magnitude of DMN effects approached or equaled the direct effects of diet for some traits, indicating that DMN effects constitute a major source of variation.

In the mitonucleogenotype analysis (Fig 3A), the effect of the diet:mitonucleogenotype interaction even exceeded that of diet for fertility in both feeding paradigms. For development, in both feeding paradigms, the effect of the diet:mitonucleogenotype interaction equaled the effect of diet. Again, this suggested that response to diet in these populations could not be understood without first accounting for mitonucleogenotype.

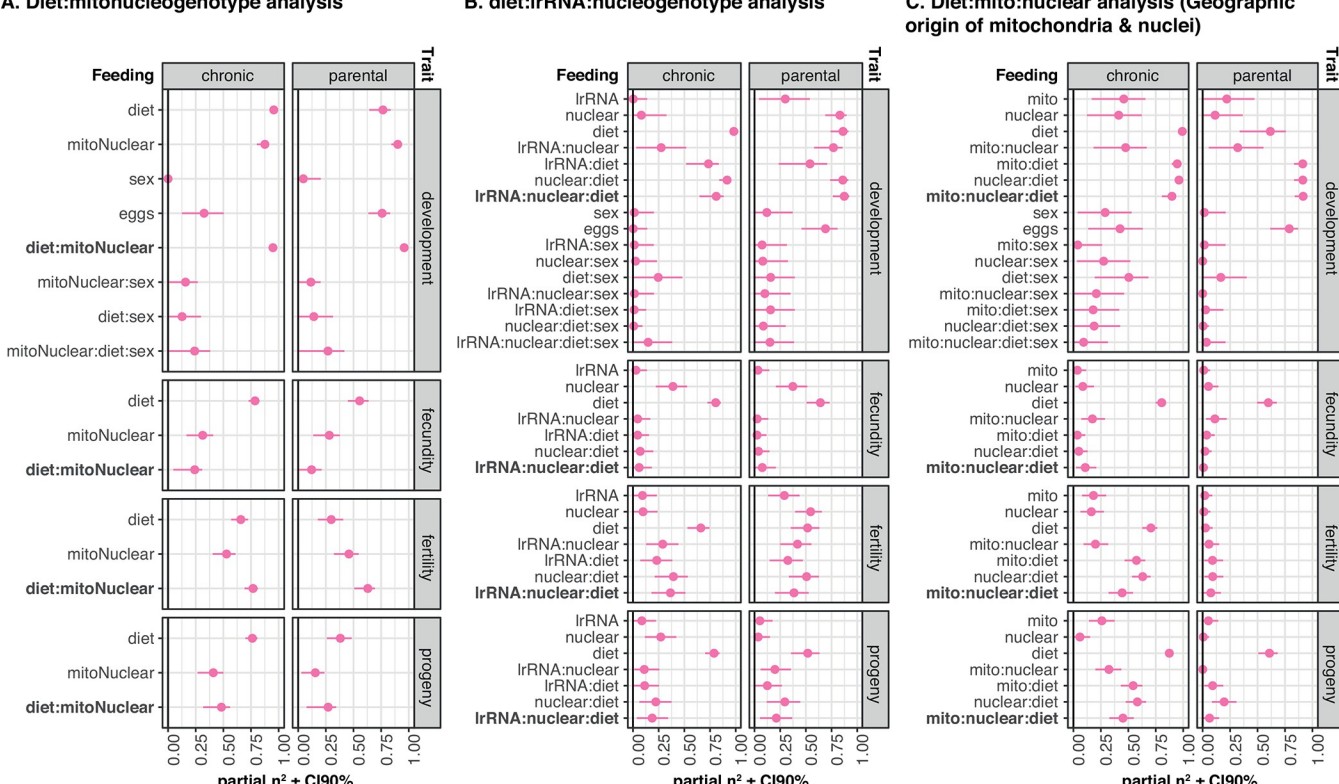

**Fig 3. Effect size calculations reveal substantial modification of response to diet by *mt:lrRNA*.** The 3 sets of panels show a standardised way of calculating the impact of terms in statistical models (effect size—partial $\eta^2$). Error bars show confidence intervals for partial $\eta^2$ estimate. Note that for some estimates, confidence intervals are not visible because error bars are smaller than the plotted point. For each set of plots, facets represent the 2 different feeding paradigms (columns) and the different traits under study (rows). Text to left of each set of columns represents model terms. Partial $\eta^2$ calculated from GLMMs (fecundity, progeny, and fertility) or Cox mixed models (development). Partial $\eta^2$ is calculated for each of the 3 approaches to analyzing the phenotype data. (**A**) Sequence-informed diet:mitonucleogenotype analysis, as per plots in Fig 2C, and statistical analysis in S6 Table. Effect size calculations show that diet: mitonucleogenotype interaction has impacts greater than or equal to main effect of diet for development and fertility in both feeding paradigms. (**B**) diet: lrRNA:nucleogenotype analysis, subset of mitonucleogenotypes highlighted in Fig 2C, and statistical analysis presented in S7 Table. Effect sizes suggest that, for progeny and fertility of these populations (mitonucleogenotypes 5–8), the variation resulting from the interaction of *lrRNA* polymorphism, nucleogenotype, and diet is equivalent to standing genetic variation from nucleogenotype and *lrRNA* polymorphism, and also an equivalent determinant of response to diet. (**C**) Geographic origins–informed diet:mito:nuclear analysis, as per S6 Fig, and statistical analysis presented in S8 Table. For all traits in the chronic feeding paradigm, diet consistently had the largest effect size, but diet-mito-nuclear effect size was either greater than or equal to mito-diet and nuclear-diet terms, and also larger than mito or nuclear main effects. In the parental feeding paradigm, for fertility and development but not fecundity or progeny, diet-mito-nuclear effect size was either greater than or equal to other genetic modifiers of response to nutrition, and by ranking greater than main mito or nuclear effects. Data underlying the graphs shown in the figure can be found in S15 Table.

The lrRNA:nucleogenotype:diet interaction was an important source of variation among mitonucleogenotypes 5, 6, 7, and 8 (Fig 3B). For fertility during chronic feeding, lrRNA: nucleogenotype:diet effects were bigger even than for nucleogenotype. Most strikingly, for development in both feeding paradigms, effect size for the lrRNA:nucleogenotype:diet interaction was large, approaching or even equal to main effects of diet. Altogether, these results suggest that epistasis between nucleogenotype and a SNP in noncoding mtDNA can dictate response to diet, which can produce more phenotypic variation than the main effects of mitochondrial or nuclear genotype and can equal the effect of diet.

In the "geographic" analysis (Fig 3C), after chronic feeding, DMN effect sizes were greater than, or equal to, mito:diet and nuclear:diet effects for egg laying, progeny, and fertility. For development in both paradigms, DMN effect sizes were large, on par with diet, diet:mitogenotype, and diet:nucleogenotype, revealing DMN interactions as a major source of variation for

developmental impacts of chronic dietary change. Fertility effects were more pronounced after chronic feeding than after parental feeding, but in both paradigms, DMN effect sizes were approximately 75% of diet's main effect, as were mito:diet and nuclear:diet terms, suggesting that these factors do not simply modulate effect of diet, but their interaction with diet is a substantial source of variation outright. In fact, in the parental nutrition paradigm, DMN effect sizes for fertility outranked the main effect of diet, although with overlapping confidence intervals and modest effect size. However, for development, effect sizes for DMN terms exceeded nearly all other terms (except for the lower-order diet:mito and diet:nuclear interactions), without overlapping confidence intervals—this exceeded even the main effect of diet, suggesting that dietary regulation of this development could not be properly understood without accounting for mitonucleogenotype.

We also validated our effect size calculations orthogonally by assessing how a range of alternative models described the data (AIC analyses) and by calculating variance explained ($r^2$) by each model, which gave congruent results (Text D in S1 Text and S9 and S10 Tables). These analyses suggest that mitonucleogenotype can modulate response to dietary variation, and the emergent interaction can produce as much phenotypic variation as the main effect of diet.

Thus, the effect of diet cannot be understood in the present panel of populations without accounting for mitonucleogenotype. A SNP in a subset of the panel of populations (populations 5 to 8) bore mitochondria differentiated only by a SNP in *lrRNA*, thereby associating variation in this mtDNA locus to diet-mito-nuclear effects. Our finding that these populations were distinguished only by this SNP, and no other, suggests that this SNP alone can be sufficient to underpin epistatic interactions with the nucleus, which can lead to distinct fitness impacts of altering diet.

## Discussion

Predicting phenotype from genotype is a long-standing challenge. To this end, genome-wide association studies (GWASs) have flourished. Two overarching findings of the era of GWAS are that nonprotein-coding variation is more important than previously expected and that additive effects of independently segregating variants do not fully explain quantitative trait variation [4,32]. This latter finding implies "missing heritability," suggesting that additional processes are at work. Two hypothetical explanations are that genotype-by-genotype epistasis (G*G) and genotype-by-environment (G*E) interactions create nonadditive effects. Speculation about epistasis has led to the "omnigenic model," which posits that variation in a given trait is likely explained by G*G between a few "core genes," and the sum effect of many (or all) small-effect variants throughout the rest of the genome [4,32]. Mitonuclear interactions may be a useful illustration of the omnigenic model, with epistasis between the few genes on the mitochondrial genome and the sum of nuclear genomic variants producing substantial phenotypic variation [7]. The omnigenic model predicts that identifying core genes may enable explanation of substantial phenotypic variance and that certain SNPs in core genes may limit or accentuate penetrance of nucleogenotype variation. Our results suggest that core genes with respect to metabolism are to be found on mtDNA and that in a subset of populations the outcome of dietary variation for specific nucleogenotypes depends on a single allele in the mitoribosomal 16s rRNA.

Why should variation in the mitoribosome affect how specific nucleogenotypes respond to specific diets? It is perhaps logical that variation in factors that affect regulatory processes like protein translation should affect penetrance of nuclear variation. *mt:lrRNA* sits high in a hierarchy of factors that control cellular function, since it encodes a structural unit (16s rRNA) of an organelle that translates proteins, and those proteins are responsible for ATP production for use by the whole cell. Furthermore, *mt:lrRNA* forms the mitoribosome in complex with

nuclear-encoded proteins, so there is clear scope for mitonuclear interactions to mediate function of the mitoribosome. Our findings join others showing that mitochondrial protein translation is a mechanistic fulcrum of mitonuclear interactions. In the copepod *Tigriopus californicus*, mitoribosomal proteins encoded by the nuclear genome show apparently compensatory evolution in response to rapid mtDNA evolution [33]. These genetic effects can also be environment dependent: In *Drosophila*, an SNP in mtDNA-encoded $tRNA^{Tyr}$ can cause male sterility, dependent on nuclear context—specifically the tyrosyl tRNA synthetase *Aatm* [24]—and this interaction is subject to thermal variation [34]. This interaction can be replicated by modifying photoperiod during development [35], which alters metabolic requirements, therefore suggesting that the epistatic effect of the tRNA variant is mediated by environment-specific energetic requirements and not by more general effects of temperature. Moreover, increasing the ratio of dietary yeast/sugar among approximately isocaloric diets can partially rescue the high-temperature sterility [15]. Since dietary yeast is the fly's source of protein, this effect could be mediated by essential amino acids, although the role of yeast as a major source of other nutrients (e.g., lipids and vitamins) [36] mean that other nutrients may have been causal [15]. The present study has examined the effect of nutrient-specific variation, which modifies both availability of specific nutrients and total calorie availability. While we were unable to detect any linear effect of calories (S5 Fig), a study designed explicitly to quantify the relationship between caloric effects and mitochondrial translation [37] may be required to discern whether the effects of the *mt*:*lrRNA* SNP are due to total energy availability or caused by qualitative differences in proportions of specific nutrients. Prior studies have tended to focus on the relationship between mitochondrial haplotype or SNP, and male fertility [15,24,38,39], and our data now suggest that parental diet can underpin diet-mito-nuclear effects in offspring fitness. We note that the traits modulated by interactions with the *mt*: *lrRNA* SNP—fertility, progeny, and offspring development time—are all potentially subject to male fertility effects, though further work is required to conclude such a connection. Overall, our results extend the repertoire of environmental manipulations (specific nutrients) and mtDNA genes (*lrRNA*) that imply connections between mitochondrial protein translation, metabolism, and fitness. Translation is a critical cellular process with systemic impacts far beyond reproductive traits; for example, modest impairment of general translational machinery can prolong organismal health into old age [40,41]. We suggest that it will be interesting and important to investigate more extensively the phenotypic space affected specifically by mitochondrial protein translation, as this may provide means to individualize therapeutic interventions. Much further work is required to elucidate the molecular, biochemical, and metabolic processes that underpin mitonuclear variation, but our work suggests that a focus on mitoribosomal function may prove illuminating.

Only 2 of the 27 mtDNA SNPs we identified were predicted to change protein coding sequence. It remains to be seen if nonprotein-coding mtDNA variation is as important as nonprotein-coding nDNA variation appears to be [4,32], though our effect size calculations (Fig 3) indicate potentially large roles. Importantly, synonymous and nonprotein-coding mtDNA variation has also been reported in latitudinal clines among wild populations [20], suggesting that this variation may be an important component of natural variation in fitness. More generally, we have added to the growing body of evidence for diet-mito-nuclear interactions [5,13,16], in the context of a literature showing that outcomes of mitonuclear epistasis are environment dependent [7,15]. Thus, altogether, DMN interactions show many of the hallmarks of a major source of phenotypic variation, and we have demonstrated this for *Drosophila* fitness traits. The Darwinian view that reproduction subjugates all other processes, and the central role of mitochondria in cellular function, suggest that these interactions may be important but underappreciated sources of variation for many further traits and not just in flies.

While we partitioned phenotypic variance to SNPs, we have not attempted systematic GWAS. In general, GWAS to test genome-wide epistasis is not tractable due to the enormous sample that would be required to maintain statistical power. For mitonuclear epistasis, testing consequences of interactions between "only" every mtDNA variant and every nuclear variant would be simpler than testing every pairwise combination of SNPs in the genome (i.e., a*b, rather than $(a+b)^2$), but an enormous sample would still be required. However, if the omnigenic hypothesis is correct [4], such a systematic approach would fail to recognize the underlying biology, which is better modeled as epistasis between a subset of core genes (i.e., mitochondrial genes) and nuclear genomic background (e.g., represented by "background" as in our study, or, alternatively, dimension reduction, pedigrees, marker loci, or pathway-level variation). If candidate mtDNA variants can be identified, methods to test their role conclusively, by mtDNA editing, are on the horizon. mtDNA editing is in its infancy [42] but would facilitate powerful tests of how mitochondria affect outputs of nuclear variation, including response to diet. An additional question raised by our study is the mechanistic role of SNPs outside of protein-coding regions on mtDNA: Are these variants regulatory? Our analyses suggest statistical associations, which would also be testable by mtDNA editing. These tools would also enable mechanistic investigation of how mtDNA variants impact mitochondrial function (e.g., respiration, proteome), their consequences for cellular processes (e.g., metabolism, epigenome), and how their impact on phenotype and response to diet varies among nuclear backgrounds. We do not dismiss the importance of coding variation, but our data suggest that noncoding variation may yet prove important.

Coevolution between mitochondrial and nuclear genomes is expected to optimise fitness. This implies that novel combinations of mitochondrial and nuclear genomes, before coevolution, would bear a fitness cost. While we have focussed mostly on sequence-informed mitonucleogenotype and its relationship to phenotype, at the same time, we can see how phenotypes parse according to geographic origins of mitochondria and nuclei. Our findings have implications for understanding variation that could emerge when novel combinations of mitochondria and nuclei arise. To date, predictions of a disadvantage to "mis-matched" mitonuclear pairings, without coevolutionary history, have received equivocal support [9,24,43–45]. In the present data, a naturally co-occurring mito-nuclear pair (*AA*) responded uniquely poorly to dietary EAAs. In the case of population $AA_3$, (mitonucleogenotype 4), EAAs were lethal. This cost was not replicated in either population bearing the constituent mitogenotype (*AB*) or nucleogenotype (*BA*), nor in another naturally co-occurring mito-nuclear pair (*BB*). However, since *AA* was a naturally co-occurring combination, with probable past coevolution, it does not seem that the detriments of this combination were caused by a novel and poorly matched pairing. These findings indicate that costs and benefits of novel mitonucleogenotypes are not necessarily straightforward functions of mito-nuclear matching or mis-matching. We suggest that novel, non-coevolved mitonucleogenotypes may be variously deleterious, beneficial, or have unforeseen costs and benefits [46].

Diet is a major source of biological variation. But the importance of genotype-by-diet variation is increasingly recognized, with genetic variation manifesting phenotypically only under certain dietary conditions and genotype-specific responses to diet [1,47]. We manipulated 2 specific nutrient classes (EAAs and lipid) normally derived from yeast in fly food, offering greater specificity than previous DMN studies. The nutrients we identify are of particular interest, because EAAs regulate many life history and health traits, while lipid consumption is associated with the pandemic of human metabolic disease [3]. We found that impacts of dietary lipid depend on mitonuclear genotype, which may be relevant to understanding variation in impacts of high-fat human diets. The high-EAA diets that we used have parallels to high-protein diets used to increase yields of livestock and human muscle mass, and our finding that

EAAs can decrease offspring quality may give pause for thought in use of these diets. We were surprised that EAA enrichment did not enhance offspring development, because we interpreted increased parental egg laying on this food to indicate parental preference, presumably in anticipation of fitness benefits. However, the discrepancy with development and fertility may indicate that EAAs function as signals of food quality as well as metabolites, which could drive deleterious outcomes when EAA levels are not representative of the composition of food that would be found in the yeasts that flies are thought to consume in nature.

An important finding of our study is that mitonucleogenotype can modify the sign of the response to dietary variation and can even result in lethality for some genotypes on EAA enriched-diet. For traits where lethality was evident, effect size calculations suggested that impacts of DMN interactions were equal in magnitude to the main effect of diet, suggesting not only that mitonucleogenotype modulates response to diet but also that DMN interactions can be major sources of variation outright. Our results thus suggest that mitonuclear incompatibility can be diet dependent. Proportion of flies surviving from egg to adult has previously been reported to be dependent on diet-mito-nuclear interactions [13]. However, in that study, it does not seem that flies experienced widespread lethality.

We have revealed a relationship between parental nutrition and mitonucleogenotype. This is a novel finding. Transient dietary alterations and metabolic disease can drive persistent molecular and phenotypic change, within and across generations [17,48]. In our study, transient parental feeding on a high-lipid diet even accelerated offspring development in *AA* mitonucleogenotypes—a surprising finding, given that we expected this diet to be largely toxic to parents and offspring. A direct role for diet in selecting embryos can be excluded in parental effects in our study because of the standardized postembryonic environment. Instead, these effects are likely explained by (A) mitonucleogenotype-specific parental allocation of development-accelerating factors, after feeding on specific diets, or (B) mitonucleogenotype-specific selection on offspring from such factors. Further investigation will be required to discriminate between these possibilities; for example, it may be illuminating to investigate variation in the metabolome of offspring whose parents had distinct mitonucleogenotypes and were fed on varied diets. More generally, after both chronic and parental feeding, effects manifested most strongly in postembryonic traits, i.e., for fertility, development time, and total progeny, and a dietary and nuclear interaction with a C/T polymorphism in *mt:lrRNA* was sufficient to cause these effects. Interestingly, in embryos, lrRNA has been localized outside the mitochondria, in polar granules, suggesting functions in germline determination [49] and highlighting this noncoding RNA as a potential mechanistic link to postembryonic variation. Additional or alternative candidate mechanisms to mediate mitonuclear variation in parental effects include altered epigenetic marks, nutrient provision from mother to offspring, or microbiota. It may be illuminating in the future to ask if effects of parental diet are modulated by parental mitonucleogenotype (e.g., differential nutrient allocation to eggs, gamete epigenome) or mitonucleogenotype-dependent processes in offspring (e.g., differential response to altered maternal nutrition). More generally, our study suggests that nonprotein-coding variation in mitochondria may modify cellular function in ways that are not yet understood but appear to depend on dietary and nuclear genetic context. The SNP in *mt:lrRNA* may, for example, modify protein translation. A role for small RNAs encoded by the mitochondrial genome is also emerging [50,51], which may, for example, modulate posttranscriptional gene regulation. Altered mitochondrial metabolism resulting from altered regulatory processes will alter overall cellular metabolism, which can have myriad downstream consequences. DMN effects on appetite, feeding rate, and nutrient allocation may be at play. Much further work is now required to elucidate these mechanisms.

In summary, our study shows that (A) specific nutrients' fitness effects are shaped by interplay of mitochondrial and nuclear genetic variants, that (B) effect sizes of DMN interactions can equal effects of diet, and that (C) a single-nucleotide substitution in mitochondrial 16S rRNA is sufficient to elicit these effects. This suggests that variation in mtDNA does not need to change the protein sequence to interact with the nuclear genome to dictate optimal nutrition.

## Materials and methods

### Flies

*D. melanogaster* populations were established as described in Fig 1A. The ancestral Australian population was isolated in Coffs Harbour, New South Wales, Australia [52]. The Benin population is the widely used *Dahomey* population, isolated in the 1970s in Dahomey (now Benin). The cytoplasmic endosymbiont *Wolbachia* was cleared by tetracycline treatment 66 generations prior to experiments. For each population, 45 females of the desired mitochondrial background were crossed to 45 males of the desired nuclear background per generation, sampling the daughters of each cross and backcrossing these to males of the desired nuclear background. Iterating this process over many generations led to introgression of the desired nuclear background (from males) into each mitochondrial background. Fly populations were maintained at 25˚C on development medium (see below) throughout their history prior to experimentation. For experiments, flies were collected upon eclosion to adulthood and fed fresh developmental medium before being assigned at random to experimental medium in groups of 5 males and 5 females at 3 to 5 days posteclosion. Experimental flies were maintained at 25˚C and transferred to fresh media every 48 to 72 hours for 1 week. Flies were transferred to fresh medium 24 hours before egg laying experiments. For development experiments, eggs were incubated at 25˚C and pupation and eclosion were scored daily. Eclosing adults were lightly $CO_2$ anaesthetised before counting and sexing.

### Diets

Our study used 2 distinct types of base media. *Drosophila* populations were constructed and sequenced while feeding on development medium. To manipulate nutrient content, experimental media (see below) were fed. To assay impacts of parental feeding on offspring, parents were returned to developmental medium after an interval of feeding on experimental medium. Using distinct base media ensured that, upon switching to experimental media, all flies were feeding on a new diet, and so any novelty effects were distributed evenly among conditions.

Development medium contained 1.4% agar and 4.5% brewer's yeast (both Gewürzmühle Brecht, Germany), 10% cornmeal and 11.1% sucrose (both Mühle Milzitz, Germany) (all w/v), 0.45% propionic acid, and 3% nipagin (v/v).

Experimental media built on published protocols [53–55]. These media contained a final concentration of 10% brewer's yeast, 5% sucrose, 1.5% agar (w/v), 3% nipagin, and 0.3% propionic acid (v/v). EAAs were purchased as powder (Sigma) and supplemented by dissolving into a 6.66× solution in ddH₂0 (pH 4.5) (final media concentrations: L-arginine 0.43 g/l, L-histidine 0.21 g/l, L-isoleucine 0.34 g/l, L-leucine 0.48 g/l, L-lysine 0.52 g/l, L-methionine 0.1 g/l, L-phenylalanine 0.26 g/l, L-threonine 0.37 g/l, L-tryptophan 0.09 g/l, L-valine 0.4 g/l). We added margarine (15% w/v, after [56]) to ensure that lipid supply was plant based, because wild fly physiology is likely influenced by their consumption of what appears to be a vegan diet [57,58] and because margarine sets in agar (in contrast to oils). Margarine (*Ja! Pflanzenmargarine* from Rewe Supermarkets, Germany; manufacturer's analysis 720 kcal/100 g; 80/100 g fat from 23/100 g saturated fatty acids, 40/100 g monounsaturated fatty acids, 17/100 g polyunsaturated

fatty acids) was briefly melted and then mixed thoroughly into the food (15% w/v). Final nutrient contents of rearing and control media were estimated using the *Drosophila* diet content calculator [59], with additional protein, lipid, and caloric content after nutrient supplements calculated according to margarine nutrient content report, and by assuming caloric equity between EAAs and protein at a caloric value of 4 calories/g (USDA). Vials contained approximately 5 ml of food and were stored at 4˚C for up to 1 week before use.

## Characterizing impact of high-lipid diet

The egg laying response to the novel high-lipid diet was characterized in an outbred population of *Benin* flies. These flies were reared on development medium until day 3 of adulthood, fed on lipid-enriched food or control food for 7 days, and then allowed to lay eggs on development medium overnight. This "switch" design was used to ensure that we measured the physiological capacity to lay eggs following diet treatment.

## Genome sequencing

Input DNA controls from a ChIP-Seq experiment were used for whole genome sequence analysis. Genomes of 2 replicates for each mitonuclear combination were sequenced. Pools of 50 adult flies were subjected to a standard native ChIP protocol [60]. The protocol included an MNase digestion step of 6 minutes at 37˚C using 15U of the enzyme (Thermo Fisher Scientific) per sample, which yielded fragments between 284 and 300 bp in length. DNA was extracted with the QIAquick PCR purification kit (Qiagen), and 100 ng genomic DNA of the unChIPped input/negative controls were used for library preparations with NEB Next Ultra DNA lib Prep kit for Illumina. Over 50 million $2 \times 75$bp (PE) reads per sample were sequenced on an Illumina Nextseq 500 platform in High Output (150 cycles) mode.

## Nuclear genome analysis

Raw FASTQ reads were trimmed and filtered to remove low-quality reads (minimum base PHRED quality of 18 and minimum read length of 50 bp) prior to mapping using *cutadapt* (version 2.4; [61]). Reads were mapped to the reference genome of *D. melanogaster* (Flybase Release 6.28) with *bwa aln* (version 0.7.12; [62]) using parameters optimized for Pool-seq data [63]. Mapped reads were filtered for proper pairing and a mapping quality of at least 20 using *samtools* (version 1.9; [64]). Duplicates were removed with *Picard* (version 2.18.11; http://broadinstitute.github.io/picard/), and sequences flanking indels were realigned with *GATK* (version 3.8.1.0; [65]). Sequencing depth was assessed using *Qualimap* (version 2.2.1; [66]) and ranged from 46 to 58× for autosomes and 22 to 28× for X chromosomes. Individual *bam* files from all samples were then combined into a single *mpileup* file using *samtools* (version 1.9; [64]). SNPs were called with the *PoolSNP* pipeline (version 1.05; https://github.com/capoony/PoolSNP) from the DrosEU project [67], which was specifically developed for SNP detection in Pool-seq data. Parameters for SNP calling were as those used and optimized by the DrosEU project (except minimum count was set to 10) as their dataset closely resembled the present one. The resulting *vcf* file was converted into a *sync* file using the python script *VCF2sync.py* from the DrosEU pipeline.

General genetic differentiation among nuclear genomes was assessed by PCA using the R package *LEA* (version 3.4.0; [68]) and by estimating admixture proportions with the R package *ConStruct* (version 1.0.4; [19]). Both approaches were based on major allele frequencies of nuclear SNPs on all major chromosome arms (2L, 2R, 3L, 3R, and X). In order to minimize the effects of LD, only SNPs at least 1 kb apart and outside regions of no recombination [69] were considered. Major allele frequencies were calculated with the python script *sync2AF.py*

from the DrosEU pipeline. Admixture proportions (K = 3) for each population were inferred by nonspatial modeling with 3 MCMC chains per run and 10,000 iterations.

## Mitochondrial genome analysis

Sequence data of mitochondrial genomes were retrieved from data of [18] who had sequenced all 27 mitonuclear populations in pools of 150 flies including a prior mitochondrial enrichment protocol. These data were reanalyzed with more stringent sequencing depth criteria. *Bam* files from this previous study were combined into a single *mpileup* file using *samtools* (version 1.9; [64]), which was then converted into a *sync* file with *Popoolation2* [70]. As for nuclear SNPs, a PCA using the R package *LEA* (version 3.4.0; [68]) was performed, and admixture proportions with the R package *ConStruct* (version 1.0.4; [19]) were estimated based on the major allele frequencies of all mitochondrial SNPs. Major allele frequencies were calculated with the python script *sync2AF.py* from the DrosEU pipeline. Admixture proportions (K = 3) for each population were inferred by nonspatial modeling with 3 MCMC chains per run and 10,000 iterations. Genetic differentiation for each mitochondrial SNP was assessed by estimating $F_{ST}$ according to [71] and Fisher's exact tests to estimate the significance of the allele frequency differences. Pairwise $F_{ST}$ and Fisher's exact tests per SNP were calculated between mitonuclear genotypes (replicates were pooled) with *Popoolation2* [70]. Resulting *P* values from the Fisher's exact tests were corrected for multiple testing using FDR correction [72], and SNPs significant at an FDR < 0.001 were considered as significantly differentiated between mitonuclear genotypes. The mitochondrial genome map (S4 Fig) was drawn by downloading the full mitochondrial genome sequence from flybase, uploading into SnapGene Viewer as a plasmid sequence, and adding annotations manually.

## Quantitative trait analysis

Phenotype data were analyzed in R 4.2.3. Fecundity, progeny, and fertility data were all analyzed per vial of 5 females and 5 males. Development indices were analyzed per fly. Fit of fecundity data to a negative binomial distribution was determined with firdistrplus::descdist and firdistrplus::fitdist. For "geographic" analysis of phenotypes, generalised linear mixed models of the form

$$y \sim diet*mitochondria*nuclear + (genotype)$$

were fit with lme4::glmer.nb (egg counts and progeny, negative binomial distribution) or lme4::glmer (fertility, binomial of progeny and egg counts); in which *diet* (control/EAA/lipid), *mitochondria* (A/B), and *nuclear* (A/B) were fixed factors, genotype was a random factor denoting fly population (*AA1*, *AA2*, *AA3*, *AB1*, *AB2*, *BA1*, etc.). Where relevant (S2 Fig), experimental replicate was also included as an additional random factor. An observation-level random effect was also included for fertility under chronic feeding to ameliorate overdispersion. ANOVA tests (type 3) were conducted with car::Anova, and post hoc analyses were applied with the functions emmeans::joint_tests, emmeans::pairs, emmeans::emmip [28]. Options for contrasts were set to orthogonal polynomials and sum-to-zero contrasts.

For geographic analysis of development, Cox mixed effects models of the form

$$y \sim diet*mitocondria*nuclear*sex + rearing\ density + (genotype)$$

were fit to the data with coxme::coxme. *Diet*, *mitochondria*, *nuclear*, and *genotype* terms were as in models of egg laying. *Rearing density* coded number of eggs laid in the vial in which the individual developed, to account for variation in rearing density. We chose to omit *vial* as a random effect from the development models because each vial had a matching egg count,

included as a fixed effect, and, therefore, including vial would have constituted redundant information that compromised modeling: Indeed, attempting to include vial as a random effect led to poor or failed model fitting. ANOVA and post hoc EMM tests were conducted as per fecundity analyses.

For analyses focussing on the subset of lines bearing only the *mt:lrRNA* SNP, the relevant subset of the data was modeled as above, replacing the "mitochondria" term with a factor denoting whether the mitochondria bore the C or T variant.

For "mitonucleogenotype" analysis, fecundity, progeny, and fertility data were analyzed with general linear models of the form

$$y \sim diet*mitonucleogenotype$$

using a negative binomial model (MASS::glm.nb) for fecundity and progeny counts and a binomial model for fertility data (stats::glm, specifying binomial error family). For "mitonucleogenotype" analysis of development, a model of the form

$$y \sim diet*mitonucleogenotype * sex + rearing\ density$$

was fit using survival::coxph. PCA of phenotype data was conducted with prcomp on scaled EMMs for each trait on each diet. $r^2$ was calculated with MuMIn::r.squaredGLMM. Akaike weights were calculated with MuMIn::dredge.

To overcome challenges in calculating effect sizes for 3-way DMN interactions, for each trait, we used EMMs and joint tests to calculate F ratios, from which a measure of effect size within the sample population (partial $\eta^2$) can be estimated [28,31,73]. Effect sizes were calculated with custom functions built around effectsize::F_to_eta2. Application of this function was necessarily specific to the type of model in question. In all cases, F statistics were taken from ANOVA tables returned by emmeans::joint_tests. For GLMMs and Cox mixed-effect models, degrees of freedom were taken from ANOVA tables returned by emmeans::joint_tests, with residual degrees of freedom calculated by df.residual for GLMMs, or taken from model output for Cox mixed-effect models. For GLMs, degrees of freedom and residual degrees of freedom were taken from stats::anova.

EMMs superimposed on plots were calculated by fitting GLMs of the form

$$y \sim diet*line$$

and calculating EMM per population per diet. A distinct model was used for EMM calculation because models used to calculate statistical effects did not return main effects for each population (e.g., coefficient for mitogenotype A, nucleogenotype A, control diet) and therefore did not show among-mitonucleogenotype replication (i.e., coefficients for each of population $AA_1$, $AA_2$, $AA_3$ on control diet). Confidence interval of development time EMM for population $AA_3$ on +EAA diet could not be properly calculated (total lethality should mean no error, giving rise to infinite estimates in our statistical analyses); therefore, CIs were excluded from the plot. Values were returned to original scale by exponentiation when emmeans returned logged values.

Difference indices were calculated from EMMs per population per diet described above. For each pairwise comparison, fold-change was calculated as

$$\frac{(Y - X)}{X}$$

where Y represented posttreatment value and X represented starting value. Absolutes of these values were then logged, re-signed, and scaled to a −1:1 scale. *P* values for difference in EMM

were calculated for each pairwise comparison using emmeans::pairs, from which FDR was returned with stats::p.adjust. Bubble plots were produced using ggplot2. Heatmap of nutrient content was plotted in R with superheat. Figures were assembled in Adobe Illustrator.

## Graphics

Outlines of *Drosophila* in Fig 1 were drawn by hand. Mitochondria in Fig 1 were edited from a wikimedia.org representation of an animal cell distributed under Creative Commons CC0 1.0 licence. Skull and crossbone graphics were sourced from openclipart.org under a Creative Commons Zero 1.0 Licence. Map in Fig 1A was drawn in the R package "maps" and edited using Adobe Illustrator.

## Supporting information

**S1 Text. Text A in S1 Text.** Novel high-lipid diet represses fecundity. Text B in S1 Text. Initial fecundity experiments: Specific nutrients sufficient for DMN variation. Text C in S1 Text. Multitrait phenotyping with chronic and parental diet manipulation: Analysis by geographic origin. Text D S1 Text. AIC and $r^2$ calculations.
(DOCX)

**S1 Fig. Variance explained by PCA of SNPs sequenced by Pool-seq in each population.** Barplots show variance explained by (**A**) nuclear SNPs and (**B**) mitochondrial SNPs. Data underlying the graphs shown in the figure can be found in S16 and S17 Tables.
(PDF)

**S2 Fig. Preliminary investigation of impacts of diet:mito:nuclear interactions in the present set of lines, and impact of a novel high-lipid diet.** (**A**) Reproductive manipulation by enriching fly medium with plant-based lipid. Egg laying by wild-type *Benin* flies (the ancestral population from which *B* populations were derived) after 7 days feeding on control medium (10% yeast, 5% sugar) and 15% added plant-based lipid source (margarine). After 1 week of feeding, flies were switched to development medium for egg-laying assay to ensure that any effects resulted from physiological impacts of manipulation and not differences in preference for oviposition on the food. Boxplots show medians, first and third quartiles, and fifth and 95th percentiles. Two-sample *t* test t = 1.98, df = 16, *p* = 0.03. Data shown per fly. (**B**) Key and experimental design. Flies were reared from egg to adult on rearing food and allocated at random to experimental media 6–48 hours after eclosion, at a density of 5 of each sex per vial. After 7 days, flies laid eggs on fresh food for 24 hours. (**C**) Mitonuclear variation in fecundity response to nutrient enrichment. Plot shows eggs laid in vial of 5 females and 5 males over 24 hours. Boxplots show medians, first and third quartiles, and fifth and 95th percentiles. Points to left of each box show raw data. Connected points to right of each box show estimated marginal means (EMMs) with 90% confidence intervals. Data shown per vial (5 females + 5 males). Egg counts are presented as *x+1* to enable plotting log values. (**D**) Fecundity does not correlate caloric content of experimental media. Scatterplots show eggs at each caloric level, with facets per each combination of mitochondrial (columns) and nuclear (rows) genotype. Diet indicated by color. Populations show smoothed spline through points. Egg counts are presented *x+1* to enable plotting log values. Trait values do not linearly correlate with calories; therefore, caloric content is no more informative than modeling diet as an unordered factor. (**E**) Technical repeatability of diet:mito:nuclear effect between replicate experiments. Each point shows mean egg laying per population per diet in each of 2 replicate experiments, with the replicates of each haplotype grouped by dashed populations. Means were correlated between experiments (Pearson's r = 0.87, *p* = $7.6 \times 10^{-12}$). (**F**) Biological repeatability of diet:

mito:nuclear effect among replicate lines. Bubble plot shows response index—signed, logged, absolute fold-change in specified comparisons of EMMs—with point size scaled to indicate probability of observed difference (−log10 FDR), and border opacity indicating threshold of statistical significance (FDR ≤ 0.05). Fold-change calculated for conditions on y-axis relative to conditions on x-axis, e.g., bottom-right cluster of points shows increase on EAA-enriched media relative to control. Points along diagonal show comparisons within replicate genotypes on the same diet, with few significant differences among replicate genotypes. In response to lipid enrichment, the same changes were always evident among replicates of the same genotype, and in response to EAA enrichment, similar changes were evident in some replicates. Boxes indicate comparisons among replicates of the same genotype on the same diet. Data underlying the graphs shown in the figure can be found in S18–S20 Tables.
(PDF)

**S3 Fig. Significantly differentiated loci between mtDNAs *A* and *B*.** In total, 146 SNPs were observed within the 12 populations, of which 27 were significantly differentiated between populations with mtDNAs of different origins. Significant allele frequency differences were assessed by Fisher's exact test (FDR < 0.001). Stacked barplots show allele frequencies at each locus, per population. Data underlying the graphs shown in the figure can be found in S21 Table.
(PDF)

**S4 Fig. Graphical representation of *Drosophila melanogaster* mitochondrial genome and significantly differentiated SNPs.** Red shows protein-coding genes; blue shows tRNAs; and purple shows rRNAs. Positions of significantly differentiated SNPs shown in orange, with position and alleles.
(PDF)

**S5 Fig. Trait values do not correlate linearly with the caloric content of experimental media.** Trait and feeding paradigm indicated above each panel of plots. Within each panel, scatterplots show trait values at each caloric level, with rows for each mitonucleogenotype. Diet indicated by color. Populations show smoothed spline through points. Egg and progeny counts are presented $x+1$ to enable plotting log values. Trait values do not linearly correlate with calories; therefore, caloric content is no more informative than modeling diet as an unordered factor. Data underlying the graphs shown in the figure can be found in S22 Table.
(PDF)

**S6 Fig. Impacts of mitonucleogenotype on response to chronic and parental nutritional change, with data parsed per population in the study.** (**A**) Key and experimental design. Flies were reared from egg to adult on rearing food and allocated at random to experimental media 6–48 hours after eclosion, at a density of 5 of each sex per vial. After 7 days, flies laid eggs on fresh food for 24 hours, followed by a further 24 hours on standardized rearing medium. (**B**) Mitonuclear variation in response to chronic and parental changes in nutrition. Panels show EMMs (±95% CI) for trait indicated on y-axis. Feeding paradigm and mitonuclear variation are indicated at the top of the plot. Colors encode diet as per panel A, egg and progeny counts are presented as $x+1$ to enable plotting on log scale. Development index shows EMMs for Cox mixed-effects models of proportion eclosed over time, excluding sex from plot. Development data are plotted in full as Kaplan–Meier plots in panels (C) and (D). Note the exclusion of EMMs for development of genotype $AA_3$ in chronic feeding: EAA lethality prevented meaningful estimation. (**C, D**) Kaplan–Meier plots of development for the indicated feeding paradigms. Plots show proportion eclosed over time. Colors encode diet as per panel

(A). Data underlying the graphs shown in the figure can be found in S23 and S24 Tables.
(PDF)

**S7 Fig. Diet-mitonucleogenotype interactions and the architecture of phenotype.** PCA shows ordination of populations according to mitogenotype, nucleogenotype, and diet. Results shown from PCA of scaled and mean-centered EMMs, split by facets per genotype, with mito-nucleogenotype split by rows. Data underlying the graphs shown in the figure can be found in S25 Table.
(PDF)

**S8 Fig. Structure of statistically significant differences among populations.** Bubble plot shows response index—signed, logged, absolute fold-change in specified comparisons of EMMs—with point size scaled to indicate probability of observed difference ($-\log 10$ FDR), and border opacity indicating threshold of statistical significance (FDR $\leq 0.05$). Fold-change calculated for conditions on y-axis relative to conditions on x-axis, e.g., bottom-right cluster of points shows increase on EAA-enriched media relative to control. Points along diagonal show comparisons within replicate genotypes on the same diet, with few significant differences among replicate genotypes. In response to lipid enrichment, the same changes were always evident in replicate genotypes, and in response to EAA enrichment, similar changes were evident in some replicates. Boxes indicate comparisons within replicate populations on the same diet. Data underlying the graphs shown in the figure can be found in S26 Table.
(PDF)

**S1 Table. Rounds of introgression at different phases of the study.**
(XLSX)

**S2 Table. Percentage nuclear admixture (most prevalent background) per population.**
(XLSX)

**S3 Table. Significantly differentiated SNPs between mtDNAs A and B.**
(XLSX)

**S4 Table. Confirmatory DMN for fecundity (GLMM ANOVA tests).**
(XLSX)

**S5 Table. Major allele frequencies of significantly differentiated SNPs between mtDNAs A and B derived from 2 different rounds of sequencing (in 2016 and 2018).**
(XLSX)

**S6 Table. ANOVA tests (Type III) of diet*mitonucleogenotype interactions (GLMs and Cox proportional hazards).**
(XLSX)

**S7 Table. ANOVA tests (Type III) of diet*lnRNA interactions (GLMs and Cox proportional hazards).**
(XLSX)

**S8 Table. ANOVA tests (Type III) of diet*mito*nuclear interactions (GLMMs and Cox mixed-effects models).**
(XLSX)

**S9 Table. AIC and $r^2$ values for permuted GLMMs of diet*mito*nuclear interactions, for fecundity, progeny, and fertility in chronic and parental feeding paradigms.**
(XLSX)

**S10 Table. AIC for permuted Cox mixed-effects models of diet\*mito\*nuclear\*sex interactions, for development in chronic and parental feeding paradigms.**
(XLSX)

**S11 Table. Source data Fig 1B.**
(XLSX)

**S12 Table. Source data Fig 1C.**
(XLSX)

**S13 Table. Source data Fig 1D.**
(XLSX)

**S14 Table. Source data Fig 1E.**
(XLSX)

**S15 Table. Source data Fig 3.**
(XLSX)

**S16 Table. Source data S1A Fig.**
(XLSX)

**S17 Table. Source data S1B Fig.**
(XLSX)

**S18 Table. Source data S2A Fig.**
(XLSX)

**S19 Table. Source data S2C–S2E Fig.**
(XLSX)

**S20 Table. Source data S2F Fig.**
(XLSX)

**S21 Table. Source data S3 Fig.**
(XLSX)

**S22 Table. Source data (fecundity, progeny, fertility) Figs 2C, 2D and S5.**
(XLSX)

**S23 Table. Source data (development, chronic feeding) Figs 2C, 2D and S6B–S6D.**
(XLSX)

**S24 Table. Source data (development, parental feeding) Figs 2C, 2D and S6B–S6D.**
(XLSX)

**S25 Table. Source data S7 Fig.**
(XLSX)

**S26 Table. Source data S8 Fig.**
(XLSX)

## Acknowledgments

We thank L. Holman, S. Parratt, C. Selman, E. Combet, D. Marcu, D. Sannino, V. Howick, and J. Rolff for helpful discussion. C. Froschauer and R. Dobler provided invaluable advice in setting up experiments and the populations.

## Author Contributions

**Conceptualization:** Adam J. Dobson, Klaus Reinhardt.

**Data curation:** Adam J. Dobson.

**Formal analysis:** Adam J. Dobson.

**Funding acquisition:** Adam J. Dobson, Klaus Reinhardt.

**Investigation:** Adam J. Dobson, Susanne Voigt, Luisa Kumpitsch, Lucas Langer, Emmely Voigt, Rita Ibrahim.

**Methodology:** Adam J. Dobson, Susanne Voigt, Damian K. Dowling, Klaus Reinhardt.

**Project administration:** Adam J. Dobson.

**Supervision:** Adam J. Dobson.

**Visualization:** Adam J. Dobson.

**Writing – original draft:** Adam J. Dobson.

**Writing – review & editing:** Adam J. Dobson, Susanne Voigt, Damian K. Dowling, Klaus Reinhardt.

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
