## [Editor Report · Decision Letter 0]

9 Aug 2022

Dear Adam, 

Thank you for submitting your revised Review Commons manuscript entitled "Diet's impact dictated by synonymous mitochondrial SNP interacting with nucleotype" for consideration as a Research Article by PLOS Biology.

Your manuscript has now been evaluated by the PLOS Biology editorial staff, as well as by an academic editor with relevant expertise, and I'm writing to let you know that we would like to send your revised submission out for re-review. Please accept my apologies for the extraordinary time that it has taken us to secure expert advice at this challenging time of year.

IMPORTANT: The Academic Editor has assessed the two existing Review Commons reviews, but (based on the expertise of the reviewers) feels that some aspects of your study have not yet been fully assessed. S/he has therefore asked us to recruit one or two new reviewers to fill this expertise gap (this is not unprecedented with Review Commons manuscripts).

Once your full submission is complete, your paper will undergo a series of checks in preparation for further peer review. After your manuscript has passed the checks it will be sent out for review. To provide the metadata for your submission, please Login to Editorial Manager (https://www.editorialmanager.com/pbiology) within two working days, i.e. by Aug 11 2022 11:59PM.

Kind regards,

Roli

Roland G Roberts, PhD

Senior Editor

PLOS Biology

rroberts@plos.org

on behalf of

Ines Alvarez-Garcia, PhD

Senior Editor

PLOS Biology

---

## [Decision Letter · Decision Letter 1]

23 Oct 2022

Dear Dr Dobson,

Thank you for your patience while your revised manuscript entitled "Diet's impact dictated by SNP in mitoribosomal 16S rRNA interacting with nucleotype" was peer-reviewed at PLOS Biology and please accept my sincere apologies for the delay in providing you with our decision. The manuscript has now been evaluated by the PLOS Biology editors, an Academic Editor with relevant expertise, one of the original reviewers from Review Commons and a new one that we recruited after advice from the Academic Editor.

As you will see, the reviewers have mixed opinions. While the original reviewer appreciates the improvements done in the manuscript during the revision and asks for several clarifications, the new reviewer raises concerns regarding the general presentation of the data, the statistics and the experimental design – particularly, the mtDNA genetics. In addition, the reviewer thinks that several statements regarding novelty are overstated and should be toned down. After discussing the reviews with the Academic Editor and the rest of the team, we think that all the issues raised by Reviewer 3 would have to be satisfactorily addressed in order for us to consider the manuscript further for publication and that we will consider it only as a Short Report (please look at our guidelines regarding this format - https://journals.plos.org/plosbiology/s/what-we-publish#loc-Research-based-content). We also think that the identification of a candidate SNP in the mtDNA is interesting and follow up experiments investigating this as a causal SNP would strengthen the manuscript.

In light of the reviews, which you will find at the end of this email, we would like to invite you to revise the work to thoroughly address the reviewers' reports.

Given the extent of revision needed, we cannot make a decision about publication until we have seen the revised manuscript and your response to the reviewers' comments. Your revised manuscript is likely to be sent for further evaluation by a subset of the reviewers.

**IMPORTANT - SUBMITTING YOUR REVISION**

3. Resubmission Checklist

a) *PLOS Data Policy*

b) *Published Peer Review*

Sincerely,

Ines

--

Ines Alvarez-Garcia, PhD

Senior Editor

PLOS Biology

Reviewers' comments

Rev. 1:

This paper investigates mitochondrial x nuclear x diet interactions in Drosophila melanogaster. They create fully-factorial combinations of mitochondrial and nuclear genomes from Australia, Benin, and Canada. These populations are then exposed to various diets, including a control diet, a diet high in essential amino acids, and a diet high in plant-based lipids. They screened for evidence of repeatable mitonuclear effects on fecundity and additional fitness traits looking for influence of mito-nucleotype on response to chronic vs parental dietary changes, which they found evidence of. However, the effect of mito-nucleotype on traits was variable. Certain mito-nucleotypes exhibited lower to near lethality progeny counts after amino acid feeding (which is thought to promote fecundity), while others exhibited normal counts. When quantifying the size of various effects, they found that mitonucleotype interactions often had comparable effect size to that of diet:mitotype interactions, diet:nucleotype interactions, and diet on its own. Finally, they were able to associate this mito-nucleotype interaction with a mt:lrRNA C/T polymorphism that had nucleotype-dependent effects on fertility.

I have previously reviewed this manuscript for a different journal, and I find the author's response to my review comments satisfactory. In the new draft, I noticed rewording of several sections, which I think improve the manuscript and will help readers understand the methods and results. I appreciate the additional commentary on the identified mt:lrRNA that housed an mtDNA SNP associated with phenotypic differences between lines.

I believe this study and the associated results will obviously be of interest to those studying the evolution of mitochondrial x nuclear x environmental interacts but also to those interested in the effects of nutrients/diet in Drosophila. I also find the comparison to the omnigenic model quite compelling, and I appreciate the idea that mitochondrial genes could contribute to the "core gene" set for several notable phenotypes - potentially through a regulatory mechanism.

I have only a few minor comments (nearly all typos/clarifications).

Page 15, line 75 - This sentence confused me - "This approach is expected to dilute with the paternal nucleotype and before eventually purging the F0 mother's nucleotype, while retaining the F0 mother's mitotype." I think the "and" is erroneous.

Page 17, line 118 - 120 - This may be a naive question. To ensure genetic consistency you use the same parents in each paradigm. These parents spend one week on experimental media (chronic paradigm) and then are allowed to lay eggs for 24 hours. They are then switched to a universal medium for another 24h of egg laying (parental paradigm). Is 24 hours on the universal medium sufficient to remove/minimize the effects of the experimental media? Could diet effects from the experimental media persist into egg laying while on the universal medium?

In my previous review, I asked about co-adaptation between the mitochondrial and nuclear genome. The response to reviewers mentions that a commentary on co-evolution between the mitochondrial/nuclear genomes was added on lines 289 - 297. These lines do not mention co-evolution/adaptation, and I could not find any commentary in the rest of the discussion. I'm curious whether the referenced text was not added to the version we have.

Rev. 3:

Major comments:

The main text should more clearly describe the underlying genetics. Was the Australia source stock/population segregating mtDNA haplotypes that are very similar to Benin? If so, why? Was a single female sampled to provide the mtDNA for each replicate mitochondrial-introgression line? If not, do you have evidence that there are not segregating mtDNAs within each replicate line? In the end, this is fortuitous because it enables a candidate SNP to be identified. However, once the reader gets to figure 2, it makes it seem inappropriate to include AA3 as an "A" mitotype in the analyses done in figure 1. Why is C included in the paper at all and what is the relationship between mitotypes in B and C populations? Figure S1 assigns the mtDNA in AA3 to Canada, but the figure 2 alignment makes the mtDNA appear to be very similar to Benin. Does the Benin/Canada-like mitotype(s) segregate in the "A" stocks used for the Canadian and Australian cytotype? In figure S1, the A nuclear genome has more admixture from C and the A mitotype appears to have C mtDNAs. Is this from past contamination of a C/B-like mtDNA into the A source population? Is the remnant C nuclear contribution to the A nuclear introgression lines what randomly remains after introgression or is it potentially selected for by the presence of the B/C mtDNA? Is this from evolutionary history in the source populations in the lab or from the history of introgression during the experiment? I write B/C-like because it was not clear from the combination of information from fig 2 sequence alignment and the fig S1 structure analyses what is the source of the AA3 mtDNA. Maybe a mtDNA haplotype map might help here?

One could imagine a more streamlined presentation of the data where the mtDNA genotyping is presented first with all subsequent analysis presented using the sequence-based mitonuclear genotypes rather than the AA,AB,BA,BB categorization.

Were these cytotypes cleared of or checked for the presence of wolbachia?

The manuscript describes mito-nuc-diet quantitative genetic variation for a number of traits related to fitness in flies. This is an important area of research and because of what appears to be a Benin-like mtDNA haplotype segregating in the Australian population (or the original lab stocks), there is the ability to identify a candidate mitochondria SNP in a mt-ribosomal RNA. This is an interesting result. As pointed out by reviewer 2 from review commons, this adds to a growing number of studies in flies that demonstrate mito-nuc-diet interactions as an important source of variation for phenotypes, including reproduction, development and survival. A strength of the design is the ability to contrast chronic versus parental diet effect, which is novel, particularly how this implicates mito-nuclear-diet interactions as important for provisioning gametes. This could be discussed more.

To treat this three-way interaction as "unprecedented" incorrectly represents the literature and the paper ultimately misses the opportunity to focus on the interesting biology of the candidate SNP that is identified. I fail to see the distinction that you make in response to reviewer #2 from reviewer commons that other studies have "focused on physiology and evolution, and we are not aware of prior studies that have shown that mitonuclear incompatibility is diet-dependent." Other studies in flies, including some that you cite, have shown diet can modify mito-nuclear incompatibilities that cause sterility and impact egg-to-adult survival. The diet manipulations in other studies are no less specific than those used in this study, just different.

The paper could be more focused on the past literature and the underlying biology -- how a mitochondrial SNP putatively affecting mitochondrial protein translation may interact with nuclear genome and the environment to affect gametogenesis and development. There is supporting research in this area. The discussion is framed around regulatory versus coding effects. lrRNA SNPs are not typically described as regulatory. The lrRNA function in the ribosome to catalyze protein synthesis. SNPs in these genes have the potential to affect the function of the molecule that they encode. The framing with the omnigenetic model also seems out of place. A single mtDNA SNP with detectable effects on fitness is not exactly the type of variation that the omnigenetic model describes. Although, clearly any GxGxE interaction indicates variation that does not have deterministic effects on fitness and thus may be a source of complex genotype-phenotype relationships.

The main text has no methods and the supplement is extensive, making it hard for the reader to find the key methodological and results needed to understand the main text.

Additional comments:

The questions in the abstract are not clear. I do not know what is meant by: "Are these "mitonuclear" effects deterministic with regard to optimal nutrition?" Very few genetic effects are deterministic; GxGxE variation is by nature not particularly deterministic, as they are genetic effects that are modifiable by environmental context. Or is the question whether mitonuclear genotype determine optimal nutrition, which I also find to be an odd question.

I fail to understand the repeatability = deterministic argument and this seems like a contrived framing of the introduction.

It was not clear what is meant by "the majority of mitotype diversity was represented by populations bearing only Australian or Beninese mitochondria." Figure S1 did not clarify this for me. What do you think is the source of the Australian population having a Benin (or Canada?)-like mtDNA?

Line 126, I suggest including why you have excluded a genotype from an analysis in the main text rather than the supplement.

Line 129-130, it is unclear whether diet is being considered as a categorical factor with caloric density as a co-factor. Are any of the diet modifications isocaloric? Another statistical question is how vial is being accommodated in the analyses. I was surprised by how low the p-values are in the tables given the means and variances in the fig 1 and 2 plots. Are individual vials in the experiment being treated as random factors such that individuals are not the unit of replication?

Line 142-144, the language of genetic replacement is out of place. The experiment involves a reciprocal set of mitonuclear genotypes made through introgression and not a design where you identified a genetic effect and then did a replacement to rescue the effect. I recommend sticking to the language of epistasis (GxG) where a particular phenotype is observed only in one combination of two-locus introgression genotypes.

Line 145-146, I fail to see the distinction being made here between some of the studies you cite and this study. Some of the studies that you cite provide evidence that mitonuclear incompatibilities interact with diet to impact reproductive traits, which is the same fitness trait in this study. Other studies (I think not cited) have shown effects of diet on egg-to-adult survival in mitonuclear genotypes.

Line 197-198, I do not understand what is meant by "because introgression maintained variation and therefore phenotypic variation could be associated with preceding sequence data."

Line 202-213 "how the clustering grouped populations (Figure 2A). This separated A mitotypes from B, validating our initial approach of encoding mitotype by geographic origin (Figures 2, 3)"; was a single female used to establish the two mtDNA's in the study? Are the populations segregating mtDNA variation? This should be more clearly explained in the results (or the methods provided before the results). The statement "these mitotypes were not nested within nucleotype" indicates that the designation of AA, AB, BA, and BB is by geography, but does not reflect genotype? I was surprised by this. Lines 211-213 sounds as if you have genotype data for each individual that was phenotyped; is this the case? Or, rather, do the replicate populations have a single mitonucleotype (one of the 8?)?

Lines 215-217, Why is this described as having a qualitative effect? Was it not statistically significant quantitative effect? The main text should provide statistical evidence for a nucleotype x C/T mtDNA genotype x diet interaction to support the conclusions that you draw.

Figure 1. If the three replicate populations are segregating different mtDNA (as I infer from the 5,6,7,8 mitotypes results section and figure 2 and S1), then I am not sure why categorizing the replicates as having A or B mitochondria is valid. It is hard to see how AA3 is such an outlier in the PC clustering given the data in plot 1B. From the phenotypes in 1B, the three replicate populations in the AA category behave similar to each other.

---

## [Decision Letter · Decision Letter 2]

24 Mar 2023

Dear Adam,

Thank you for your patience while we considered your revised manuscript entitled "Mitonuclear interactions dictate both direct and parental effects of diet on fitness, and involve a SNP in mitoribosomal 16s rRNA" for consideration as a Short Reports at PLOS Biology. Please also accept again my sincere apologies for the delay in providing you with our decision. Your revised study has now been evaluated by the PLOS Biology editors, the Academic Editor and one of the original reviewers. 

The review is attached below. We are pleased to offer you the opportunity to address the remaining points raised by the reviewer in a revision that we anticipate should not take you very long. We will then assess your revised manuscript and your response to the reviewer' comments with our Academic Editor aiming to avoid further rounds of peer-review, although might need to consult with the reviewers, depending on the nature of the revisions.

**IMPORTANT - SUBMITTING YOUR REVISION**

3. Resubmission Checklist

a) *PLOS Data Policy*

Please note that as a condition of publication PLOS' data policy (http://journals.plos.org/plosbiology/s/data-availability) requires that you make available all data used to draw the conclusions arrived at in your manuscript. If you have not already done so, you must include any data used in your manuscript either in appropriate repositories, within the body of the manuscript, or as supporting information (N.B. this includes any numerical values that were used to generate graphs, histograms etc.).

We need you to provide the data underlying the graphs shown in the following figures:

Fig. 1B-E; Fig. 2C, D; Fig. 3A-C; Fig. S1A, B; Fig. S2A, C-F; Fig. S3; Fig. S5; Fig. S6B-D and Fig. S7

Please also indicate in each figure legend WHERE THE DATA CAN BE FOUND. For an example see here: http://www.plosbiology.org/article/info%3Adoi%2F10.1371%2Fjournal.pbio.1001908#s5

b) *Published Peer Review*

Sincerely,

Ines

--

Ines Alvarez-Garcia, PhD

Senior Editor

PLOS Biology

Reviewers' comments

Rev. 3:

I appreciate the extensive revision which has improved the presentation of and made more clear aspects of the experimental design and results. Below are my detailed comments,

1) In the response to review, the authors indicate that they have downplayed determinism given the quantitative genetic nature of the variation they are describing. However, the title uses the term "dictate," which is synonymous with and perhaps even stronger than the term determine. Also, the word "determinant" is used throughout the paper including the first and last sentences of the introduction. At a minimum, can the authors please use terms more appropriate to the type of complex genetic effects that they are describing such as "impact," "affect" or "underlie"? 

2) Showing that mito-nuc-diet effects are on the same order of effect size is an interesting result that I think should be highlighted in the paper. This conclusion rests on the use of partial eta-squared statistics, which I understand to be SS(of focal effect) divided by the SS(error and all other factors). In Figure 3B and C, which I think are the stronger analyses for evidencing DMN effects in the study, how can so many of the factors have a partial eta-squared statistic so close to a value of one? This is particularly evident in the analysis of development. It seems to me like the statistics should sum to something on the order of 1. The other piece of information to report in the main text along with Figure 3 would be the percent of the total variation explained by the model, which would complement the point being made about equivalent relative effect sizes while also providing the reader with a sense of how much total variation is explained by diet, genetic compartments, culture/vial effects, and interactions between factors. 

Also with respect to development, how is the "index" calculated and why was development time not used as the measure? The methods indicate that every individual in a vial is being treated as an individual data point for development index? This is fine, but to avoid pseudo replication, a random vial effect needs to be included in the model to account for non-independence of individuals developing within the same culture vial. 

The data seem not normally distributed with one DMN combination in each of the two feeding paradigms to be strong outlier. To what extent are the results in Fig 3 robust to versus driven by these particular combinations? These data are not included in some statistical models, but in the main text it would be good to summarize what patterns in Fig 3 are robust to versus driven by inclusion of particular genotypes in the statistical models.

Does it not cause issues with the statistical models to have redundant information within mtDNA, nuclear genome, and the random factor of genotype?

2) The last line in the abstract refers to the mt variant as a synonymous variant. It is also presented in figure 1 as a synonymous/non-coding variant. In the response to review, the authors acknowledge that the change in the rRNA is likely functional. Yet, this remains described as a synonymous or non-coding change in the abstract and at several place in the revised manuscript, including important arguments in the discussion. The synonymous/non-synonymous categorization is used to describe nucleotide changes in protein-coding genes. Non-coding is generally used to describe parts of the genome that do not encode genes (e.g., intergenic and potentially regulatory). tRNA and rRNA genes encode functional molecules, and, unlike in a protein-coding gene, we cannot assume that any variants are synonymous. Can the authors just call it a SNP in the lrRNA when they describe it and in figure 1, change the black highlighting to be nonsynonymous/RNA-encoding? I suggest updating the last line of the introduction. Also, on Line 176, this should be non-protein coding. For line 394 of discussion ... I think this non-coding variation is different than SNPs in ribosomal RNA genes; the non-coding in this sentence is referring to the observation that there is much regulatory variation such as in intergenic regions of the genome that may have small RNAs or binding sites for regulatory factors or influence chromatin structure. Maybe a strict definition on "coding" means parts of the genome that use the genetic code to make proteins, but RNA genes are genic and encode functional RNA products that are distinct from the inter-genic, non-coding and presumably regulatory parts of the genome. Lines 449-453 of discussion should be reconsidered in light of this.

3) Lines 56-61; I would remove question "A." The authors can highlight this result from this study in the results and discussion, but the prior studies cited in line 55 use designs that address this question.

4) While I understand what the authors intend by using nucleotype (to contrast with mitotype), I suggest using the term nuclear genome. My understanding of the term nucleotype is that it is used to describe ploidy variation in the nucleus (https://doi.org/10.1086/700636;
https://pubmed.ncbi.nlm.nih.gov/6360135/). Mitonucleotypes could then be referred to as mitonuclear genotypes or genomes, which would be more consistent with the literature.

5) In the original manuscript's presentation of the experimental design, it was not clear that the lines were genetically diverse, so I apologize that this question comes upon review of the revised manuscript. Why were genetically diverse lines (although presumably these were still from lab culture, so not as diverse as intentionally outbred populations) used? Or is the genetic variation just the residual heterozygosity that we expect in lab cultures of flies? Lines 95-96 state that "the crossing scheme was designed to produce distinct mitochondrial backgrounds bearing equivalent pools of standing nuclear variation." What is meant by "equivalent pools of standing nuclear variation?" Do you have data indicating that the A, B, and C nuclear genotypes have the same heterozygosity? What is the motivation for incorporating this genetic variation?

In figure 2 there are many genotype-diet combinations that are producing values of zero (and I think without error bars or very little variance). If the mito-nuclear genotypes are segregating in the introgressed populations and these genotypes are associated with the phenotype, shouldn't you also sample individuals with combinations of genotypes that are non-sterile or non-lethal? I understand that the mtDNA variants are primarily, although not completely, fixed differences between populations, but if there is backcrossing to the outbred nuclear background, shouldn't you expect interacting nuclear factors to be segregating in the pool of individuals that you phenotype? Otherwise, you would need to posit that nuclear factors associated with DMN effects are fixed differences between populations? At a minimum, this should be clearly explained for readers.

6) A characterized "Dahomey" mtDNA has diet-dependent effects on fitness in D. melanogaster (DOI: 10.1371/journal.pgen.1007735) and a male-sterile Dahomey mtDNA has been well characterized by co-authors on this manuscript. What is the relationship of the Benin mtDNA major alleles in this study and these characterized Dahomey mtDNA variants?

7) Line 207 indicates that there were eight distinct mitonuclear genome combinations which are labeled in Figure 1G. The mtDNA for mitonuc combination 4 is distinct at a number of positions from both other "orange" and other "purple" labeled mtDNA. It is also only present in the "orange" nuclear background. Divergent diet-dependent phenotypes between groups 2 and 3 will show that mtDNA-diet effects are conditional on the nuclear background. Divergent diet-dependent phenotypes between groups 7 and 8 will also show that mtDNA-diet effects are conditional on the nuclear background. But, it seems that the SNP that is highlighted in the manuscript is being analyzed for effects using contrasts between groups 5,6,7 and 8. This took me quite a while to piece together going back and forth between figures 1 and 2. It would be helpful for readers to clearly present what contrasts are being used for what inference in the results section. For example, mito-nuclear genotypes 1 and 4 are being used for what inference? And is a DMN interaction specifically inferred as a diet-dependent genetic interaction using the subset of genotypes 5-8? I think this is the best way to define a DMN in this experimental design. Could patterns that support DMN effects be highlighted in figure 2 somehow? I found throughout the results section that DMN was used broadly and it was difficult for me to follow exactly what genetic-by-diet effects were being used to infer DMN effects. For example, both "diet:mitonucleotype" and "diet:mt:nuc" seem to be used equivalently as evidence of "DMNs" but I think only the latter is formally testing for diet by mtDNA by nuclear effects; if otherwise, then it should be clarified.

8) Figure 2. Are the p-values reported in Fig 2 multiple-test corrected?

9) Lines 241-247; also to clarify another reviewer's question, are parents developed on the different diets or only exposed as adults to the different diets before they lay eggs on the experimental and then standardized diets? Clarity on this allows the reader to infer if the effect of diet on parents and their progeny is potentially via effects on the development of parent gonadal tissue and/or via effects of diet on gametogenesis in parents. Is the effect of chronic versus parental diet confounded with the parental age at which an offspring was laid? I am willing to believe that effects are likely dietary, but good to mention this as a caveat.

10) In the results section, include statistical evidence (p-value, preferably multiple-test corrected) for statements such as in line 259 "mitonucleotype determined magnitude," and for the diet-by-mitonucleotype interaction in the GLM. Similar in line 262. Line 328-329 is where it would be important to report p-values for the mt-by-nuclear-by-diet interaction using the subset of genotypes 5-8.

11) Multiple places in the manuscript report that the C/T SNP is "sufficient to cause" DMN variation. The SNP seems "associated with" or "involved in" DMN interactions. With an epistatic mutation, one SNP alone is not sufficient; it seems by definition to also require variation in the nuclear genome and in dietary environment. This is more of an association study so the language of association seems more appropriate.

12) A final point is that I really don't know what to make of that AA3 mtDNA. Are the authors not perplexed by how this mtDNA from a putatively Australian lineage/culture in the lab appears recombinant with the Benin mtDNA sequence? In the geographic analysis, is it considered an Australian mtDNA even though the tree in Fig 2G groups it with the Benin mtDNA? Are the statistical results and conclusions robust to leaving this mtDNA out of the geographic analyses of if you consider this to be a Benin allele? I am really curious because it differs from Australia at many sites, and in each case it has the Benin allelic state, which is not what I would expect.

Minor comments:

Lines 585-587; this description of the diet manipulation was unclear.

Methods; I have not done a ChIP-seq experiment, so was unfamiliar how sequence data from a ChIP-seq experiment can be used for estimating allele frequencies as in a pooled-sequencing design. Readers may benefit from more details on this and whether the resulting coverage is within ranges that are considered good for estimating allele frequencies.

---

## [Editor Report · Decision Letter 3]

28 Jun 2023

Dear Dr Dobson,

Thank you for the submission of your revised Short Report entitled "Mitonuclear interactions define both direct and parental effects of diet on fitness, and involve a SNP in mitoribosomal 16s rRNA" for publication in PLOS Biology. On behalf of my colleagues and the Academic Editor, Nick Lane, I am delighted to say that we can in principle accept your manuscript for publication, provided you address any remaining formatting and reporting issues. These will be detailed in an email you should receive within 2-3 business days from our colleagues in the journal operations team; no action is required from you until then. Please note that we will not be able to formally accept your manuscript and schedule it for publication until you have completed any requested changes.

PRESS

Sincerely, 

Ines

--

Ines Alvarez-Garcia, PhD

Senior Editor

PLOS Biology
